

# A long-term study of aerosol-cloud interactions and their radiative effect at a mid latitude continental site using ground-based measurements

Elisa T. Sena[1], Allison McComiskey[2], and Graham Feingold[3]

[1]Institute of Physics, University of São Paulo, São Paulo, Brazil
[2]NOAA Global Monitoring Division, Boulder, CO
[3]NOAA Chemical Sciences Division, Boulder, CO

*Correspondence to*: Elisa T. Sena (elisats@if.usp.br)

**Abstract.** Empirical estimates of the microphysical response of cloud droplet size distribution to aerosol perturbations are
commonly used to constrain aerosol-cloud interactions in climate models. Instead of empirical microphysical estimates, here
macroscopic variables are analyzed to address the influences of aerosol particles and meteorological descriptors on
instantaneous cloud albedo and radiative effect of shallow liquid water clouds. Long-term ground-based measurements from
the Atmospheric Radiation Measurement (ARM) Program over the Southern Great Plains are used. A broad statistical
analysis was performed on 14-years of coincident measurements of low clouds, aerosol and meteorological properties. Two
cases representing conflicting results regarding the relationship between the aerosol and the cloud radiative effect were
selected and studied in greater detail. Microphysical estimates are shown to be very uncertain and to depend strongly on the
methodology, retrieval technique, and averaging scale. For this continental site, the results indicate that the influence of
aerosol on shallow cloud radiative effect and albedo is weak and that macroscopic cloud properties and dynamics play a
much larger role in determining the instantaneous cloud radiative effect compared to microphysical effects.

**1 Introduction**

Clouds are major contributors to global reflectivity (Trenberth et al., 2009). Thus, changes in cloud albedo, coverage, and
lifetime have a large impact on the Earth's radiation budget. Additionally, changes in precipitation patterns may have a large
impact on agriculture, the environment, and human well-being.

The influence of aerosol on clouds and its contribution to cloud radiative forcing has become a theme of much debate in the
scientific community (Boucher et al., 2013). The processes involved in cloud development, aerosol and cloud lifecycles, and
cloud radiative responses are complex and not well represented in global climate models (GCMs). Microphysical responses
associated with aerosol effects on cloud albedo follow the sequence of more aerosol resulting in more cloud condensation
nuclei (CCN), and all else equal, smaller cloud drops and a more reflective cloud (Twomey 1974, 1977). However, aerosol,
dynamics and macroscopic cloud properties are interconnected, and may result in mutually compensating effects and



adjustments that are not fully understood (Stevens and Feingold, 2009). For example, smaller drops may suppress precipitation and increase cloudiness (Albrecht, 1989) or, by enhancing entrainment and evaporation, decrease cloud amount (Wang et al. 2003; Ackerman et al. 2004; Small et al. 2009). Absorbing aerosol could also modify the atmospheric temperature profile and stability, and reduce cloud amount via the semi-direct effect (e.g., Koren et al., 2008).

Therefore, cloud microphysical variations do not necessarily manifest as changes in cloud albedo and radiative forcing (Han et al., 1998). The influence of meteorology on aerosol-cloud interaction assessments is increasingly being brought into focus (e.g., Engstrom and Eckman, 2010, Kaufman, et al., 2005, Koren et al., 2012, Chen et al., 2014, Chen et al., 2015). However, untangling the cloud microphysical effects from dynamics and isolating their contributions to the radiative balance still remains a big challenge. Direct, independent, and collocated measurements of each pertinent variable are required for

understanding the impact of the anthropogenic aerosol on the cloud radiative effect (McComiskey and Feingold, 2012). Evidence for anthropogenic aerosol influence on cloud droplet number concentration and effective radius is commonly seen from in situ airborne measurements (e.g., Warner and Twomey, 1967, Eagan et al., 1974, Ackerman et al., 2000, Twohy et al., 2005). Over the past two decades, satellite remote sensing has been widely used to study aerosol-cloud interactions over large areas (e.g., Nakajima et al., 2001, Bréon et al., 2002, Quaas et al., 2008, Costantino and Bréon, 2010), usually showing

weaker responses than airborne-based studies. Space-borne assessments of aerosol-cloud interactions face many challenges, such as cloud contamination of the aerosol measurement, aerosol humidification effects near clouds, and the difficulty in obtaining collocated aerosol and cloud measurements. Different observational scales and platforms result in large variations in the aerosol-cloud interaction assessments (McComiskey and Feingold, 2012).

The Department of Energy's (DOE) Atmospheric Radiation Measurement (ARM) Program continuously operates permanent

and mobile facilities that allow monitoring and studying the atmosphere at different sites. The unrivaled combination of in-situ and ground-based remote sensing instruments provides collocated and simultaneous measurements of different cloud, aerosol and meteorological properties. ARM ground-based instrumentation has been previously used to study aerosol-cloud interactions at several sites around the world (e.g., Feingold et al., 2003, Kim et al., 2003, Kim et al., 2008, McComiskey et al., 2009, Garrett et al., 2004). These studies focused on the microphysical aspect of aerosol-cloud interaction, analyzing a

handful, to months, to up to three years of measurements. The ARM Program has been operating at the Southern Great Plains (SGP), Oklahoma, for more than two decades (since 1992). The availability of such a large and comprehensive dataset provides an excellent opportunity to pursue a long-term study of the effects of aerosol and meteorology on clouds.

In this work, 14-years of ARM ground-based measurements at the SGP were analyzed to investigate the effects of aerosol and meteorology on clouds. Instead of quantifying the usual metrics for microphysical response to an aerosol perturbation,

we focus on the analysis of aerosol associations with cloud macroscopic variables and radiative properties. These quantities are more closely related to the cloud radiative effect and therefore represent a pragmatic pathway towards quantification.

The structure of the paper is as follows: Section 2 describes the methodology. A climatology of low, warm, non-precipitating clouds at the SGP is then presented (Section 3.1). Some simple approximations are used to illustrate the theoretical basis behind the data analysis (Section 3.2). A broad statistical analysis of more than a decade of coincident ground-based





measurements of cloud radiative properties and their relationship with meteorology and aerosol concentration is shown (Section 3.3). Two interesting cases are selected and studied more deeply to improve our understanding of the problem (Section 3.4). Common features observed in the case studies are further explored (Section 3.5). We summarize our results in Section 4.

**2 Methodology**

Coincident ground-based remote sensing and in-situ measurements of clouds, aerosol and meteorological properties from Atmospheric Radiation Measurement (ARM) deployments at the Southern Great Plains (SGP), Central Facility, near Lamont, Oklahoma (36.61ºN, 97.48ºW), were used. The period of data analysis ranges from 1997 to 2010 and includes all available data that present coincident measurements of the variables considered, subject to the restrictions described below.

The Active Remotely Sensed Cloud Locations (ARSCL) Value-Added Product (Clothiaux et al., 2000) was used to select low, warm, non-precipitating clouds from the full 14 years of data. This product combines measurements from a Ka-band cloud radar (35 GHz or 8.6 mm wavelength), a ceilometer at a wavelength of 910 nm, and a Micropulse Lidar (MPL) at 532 nm to provide, among other variables, best estimates of cloud boundaries at 10-second resolution. To avoid ice, the cloud base height $h_{CB}$ was limited between 300 m and 2000 m and the cloud top $h_{CT}$ was limited to 3000 m. Cases that presented

more than one layer of cloud were excluded from the analysis. Drizzle was avoided by limiting the maximum column radar reflectivity ($Z$) to less than -17 dBZ (Frisch et al., 1995).

Surface broadband radiative fluxes were used to obtain cloud optical depth $\tau_c$, (a parameter closely related to cloud albedo, $A_c$), cloud fraction $f_c$, and the instantaneous relative cloud radiative forcing rCRE, using the Radiative Flux Analysis Evaluation Product (RFA, Barnard and Long, 2004, Long and Ackerman, 2000, Long and Shi, 2006, Long et al., 2006).

Overcast conditions ($f_c > 0.95$) are required to retrieve $\tau_c$. $A_c$ and $f_c$ were simultaneously retrieved using piecewise polynomial fits to functions of shortwave upward and downward radiation fluxes (Liu et al., 2011; Xie and Liu, 2013). rCRE, a non-dimensional measure of instantaneous cloud radiative forcing, or cloud radiative effect (Betts and Virtebo, 2005) is defined as

$$rCRE = 1 - \frac{F_{all}^{dn}}{F_{clr}^{dn}}, \tag{1}$$

where $F_{all}^{dn}$ and $F_{clr}^{dn}$ are the broadband all-sky and clear-sky surface downwelling shortwave radiative fluxes (from 0.3 to 3.0

25   µm), respectively. The use of downwelling fluxes as opposed to net fluxes minimizes the effects of surface albedo on rCRE (Vavrus, 2006).

The aerosol index $A_i$ was calculated from the surface scattering coefficient at 550 nm $\sigma_{550nm}$ multiplied by the Ångström exponent (Å) and used as a proxy for CCN concentration (Nakajima et al., 2001)

$$A_i = \sigma_{550nm}\text{Å}, \tag{2}$$





where $Å$ and $\sigma_{550nm}$ were measured by a 3-channel nephelometer (at 450, 550 and 700 nm) at 1-minute resolution (Sheridan et al., 2001). An impactor at the inlet connected to the nephelometer alternates the cut size from 1 to 10 μm every 6 minutes. Only measurements obtained at the 1-μm size cut were selected. The data were interpolated to 1-minute resolution, when necessary. The decision to use surface measurements is both pragmatic (they are available) but also supported by the result

that at SGP the relationship between surface aerosol measurements and cloud level aerosol measurements has been shown to be uncorrelated with the degree of boundary layer vertical mixing (Delle Monache et al. 2004). Further analysis presented below confirms that our main results are only weakly dependent on the degree of mixing (Fig. 6).

Liquid water path (LWP) retrievals from a 2-channel (23.8 and 31.4 GHz) microwave radiometer (MWR) at 20-second resolution (Turner et al., 2007a) were used. Two different LWP ranges were selected. In the first part of this work (Section

3.3), our goal is to understand how several different properties impact rCRE. For that part of the study, LWP is limited between 30 and 250 g m$^{-2}$, allowing us to include cloud types ranging from low liquid water clouds (Vogelmann et al, 2012, Turner et al., 2007b), some of which are likely broken, to thicker, possibly drizzling clouds. The lower limit was set taking into account the large uncertainty in the MWR retrieval for low LWP. For the remaining analysis LWP was further restricted from 50 to 150 g m$^{-2}$. This larger restriction to the upper range was applied to minimize contribution from precipitating

events. The increased lower limit avoids very thin or broken clouds where the uncertainty in measuring LWP is high (Turner et al., 2007b).

Turbulence plays an important role in determining the number concentration of aerosol particles that are activated to become cloud droplets (e.g., Twomey, 1959, Feingold et al., 2003). The vertical component of the turbulent kinetic energy provides an estimate of the strength of the turbulent fluxes acting at cloud base. Doppler radar vertical velocities were used to

calculate a proxy for turbulence given by $w'^2 = [w-w_0]^2$, where $w$ is the Doppler radar vertical velocity at the cloud base, and $w_0$ is the average vertical velocity at the cloud base centered ± 30 min around each measurement.

The decoupling index $D_i$ is an indicator of how well-mixed the atmosphere is, and therefore how well ground-based measurements of conserved variables and aerosol properties represent the same at cloud base:

$$D_i = \frac{h_{CB} - LCL}{h_{CB}}, \qquad (3)$$

where the lifting condensation level (LCL) is calculated using ground-based meteorological measurements of surface

pressure, vapor mixing ratio, and temperature. As $D_i$ retrieval depends on $h_{CB}$ it can only be calculated in the presence of a cloud. This means that $D_i$ does not necessarily reflect the mean mixing state, unless $f_c$ is high. In broken cloud scenes, a cloud element may be well coupled, whereas the average for the entire boundary layer may be poorly coupled. This should be kept in mind in subsequent discussion.

The lower tropospheric stability (LTS), given by the difference between potential temperatures at 700 hPa and at the surface,

was also analyzed. This variable is related to the strength of the capping inversion. Studies show that LTS correlates well with the $f_c$ of low stratiform clouds (Klein and Hartmann, 1993). The potential temperatures were obtained from the Merged Sounding Value-Added Product (Troyan, 2012), version 1. This product combines radiosondes, MWR, surface



measurements and the European Centre for Medium Range Weather Forecasts (ECMWF) model output to provide several relevant meteorological parameters at 1-minute resolution, at 266 pressure levels, up to 20 km.

A summary of the instruments, the temporal resolution in the original data set, measurements and retrievals used in this work is shown in Table 1. All of the relevant variables were averaged (or interpolated, in case of $A_i$) to 1-minute resolution for the analyses presented here.

## 3 Results

### 3.1 Database characterization

A statistical analysis of the data set used in this study is performed. Relative frequency histograms show the distribution of some of the key properties that satisfied the selection criteria explained in the previous section (Fig. 1). Red bars represent the distribution obtained when LWP is limited between 30 and 250 g m$^{-2}$; the blue bars are obtained by limiting LWP between 50 and 150 g m$^{-2}$. The mean (dot), median (cross) and standard deviation (vertical lines) are shown above each distribution. The data set represents about 39,000 valid observations for the first criterion (blue) and about 66,000 for the second criterion (red). Due to the long duration of this study period, these distributions can be regarded as representative of low-level, warm, non-precipitating clouds at the SGP for the selection criteria stated above.

Figure 1a shows that the data are dominated by clouds with lower LWP, with the number of observations decreasing as LWP increases. The more restrictive LWP limit (blue bars) shows a higher relative frequency than the less restrictive limit (red bars), due to the smaller number of observations. The non-cloud properties are barely affected by changing the LWP limits. For $A_i$, $D_i$, LTS and $w'^2$ (Fig. 1i-l) the red and blue distributions are essentially the same. On the other hand, the distributions of most of the cloud properties are modified depending on the LWP limit considered. $A_c$, cloud thickness, $\tau_c$, rCRE and $f_c$ show a narrower distribution when the LWP range is restricted (Fig. 1c-f), indicating that these variables are closely related to LWP (Turner el al., 2007b).

Due to our selection criteria (low, warm, non-precipitating clouds), most of the data represent stratiform clouds, characterized by high $f_c$. Figure 1b shows that about 92% of the observations were acquired in overcast conditions ($f_c$ greater than 0.9). The number of broken-cloud observations ($f_c < 0.9$) are about 6800 and 3300 for the less and more restrictive LWP range, respectively.

To a good approximation, rCRE is directly proportional to both $A_c$ and $f_c$ (Xie and Liu, 2013):

$$rCRE \sim f_c A_c. \tag{4}$$

As most of the observations were obtained in overcast conditions (Fig. 1b), rCRE in this study is mostly determined by $A_c$, and therefore the shapes of the distributions of rCRE and $A_c$ (Fig. 1c-d) are very similar (slightly negatively skewed). Due to the polynomial criterion used to calculate $A_c$, about 0.5% of the observations resulted in $A_c = 0$. The median values obtained





for rCRE, $A_c$ and $\tau_c$ (Fig. 1c-e) were about 0.68, 0.62 and 17, respectively, for the more restrictive LWP range, and about 2 to 3% smaller when the LWP restriction was relaxed.

As expected, the $A_i$ distribution (Fig. 1i) is positively skewed indicating the predominance of clean cases (low $A_i$) over polluted cases. The distribution of the turbulence proxy ($w'^2$) peaks at 0 and rapidly decreases as $w'^2$ increases. This is due to the small number of cumulus observations in the database, which are usually associated with higher turbulent fluxes. For about one-third of the observations, $w'^2$ is higher than 0.1.

Most of the selected clouds can be classified as thin clouds (Fig. 1f). About 54% of the observations correspond to clouds thinner than 500 m, with cloud thickness peaking at about 300 m. Almost 70% of the cases correspond to clouds with $h_{CB}$ lower than 1 km, and for more than 82% of the cases, $h_{CT}$ is lower than 2 km.

By definition (Eq. 3) a value of $D_i = 0$ represents a well-mixed boundary layer whereas values greater than 0 represent progressively more decoupled boundary layers and therefore progressively weaker vertical mixing. The median of the $D_i$ distribution (Fig. 1k) is about 0.37, and about 31% of the observations show significant decoupling with $D_i$ larger than 0.5. The few cases of negative $D_i$ shown in this distribution are most likely attributed to incorrect retrievals of the $h_{CB}$. The LTS distribution (Fig. 1l) is roughly symmetrical and varies between 9 and 20 K, within one standard deviation. These values are smaller than a previously published long-term evaluation (2001 – 2010) that reported a mean value of 20.81 K for stratiform cloud LTS at SGP (Ghate et al., 2015), based on 83 radiosonde soundings obtained between 2001 and 2010, for both, nighttime and daytime. A low bias in the LTS from the merged sonde product can be expected because of the inherent smoothing of the merged soundings used in this work.

Notwithstanding the important role of $f_c$ in cloud radiative effect (Eq. 4), the predominance of high $f_c$ in this data set shifts our attention in the following analysis to the relationships amongst rCRE, $A_c$, $\tau_c$, LWP, and $A_i$.

## 3.2 Theoretical basis

For high $f_c$ conditions, cloud liquid water is an important driver of variability in cloud radiative effect because it is so tightly correlated with $\tau_c$ and $A_c$ (e.g., Kim et al. 2003; Han et al., 1998, Chen et al., 2014). Thus, we are particularly interested in the relationship between rCRE and LWP and, by contrast, the relationship between rCRE and aerosol. To give us some insight into the expected behavior of this function, a simple theoretical relation is derived.

The rCRE (Eq. 1), can be expressed as

$$rCRE = 1 - T, \tag{5}$$

where $T$ is the total cloud transmissivity.

Considering conservative cloud scattering (that is, no absorption), $T$ is obtained using a two-stream radiative transfer approximation (Bohren, 1987) given by:

$$T = \frac{2\cos\theta_0}{2 + (1 - g)\tau_c}, \tag{6}$$



where $g$ represents the asymmetry parameter of the cloud droplets and $\theta_0$ is the solar zenith angle. This same two-stream approximation yields

$$A_c = \frac{(1-g)\tau_c}{2+(1-g)\tau_c}. \tag{7}$$

Replacing $T$ (Eq. 6) in Equation 5 and performing some algebraic manipulations, the rCRE can be expressed as a function of $\tau_c$:

$$rCRE = \left[1 + \frac{2cos\theta_0}{(1-g)\tau_c}\right]^{-1}. \tag{8}$$

Equation 8 shows that, for fixed illumination angle and cloud scattering geometry, rCRE increases with $\tau_c$.

In the adiabatic regime, $\tau_c$ relates to cloud droplet concentration ($N_d$) and LWP through (Boers and Mitchell, 1994)

$$\tau_c = c(T,p)N_d^{\frac{1}{3}}LWP^{\frac{5}{6}}, \tag{9}$$

where $c(T,p)$ is a known function of temperature $T$ and pressure $p$. According to Eq. 9, the LWP contribution to $\tau_c$ is, in a relative sense, 2.5 times that of $N_d$. The same can be shown to be true for sub-adiabatic clouds (Boers and Mitchell 1994). Note that in presenting these equations with respect to $N_d$ we inherently assume a proportionality between $N_d$ and aerosol

concentration $N_a$ (or proxy such as $A_i$). If $\tau_c$ were to be cast in terms of $N_a$, the power law dependence of $\tau_c$ on $N_a$ would be less than 1/3. Because of the uncertainty in the relationship between $N_d$ and $N_a$ we use $N_d$ to simplify the theoretical arguments.

$\tau_c$ (and therefore $A_c$) thus subsumes both the amount of condensed water, a macroscale property, as well as drop (or aerosol) concentration, a microphysical property. Thus the extent to which the rCRE dependence on LWP differs for different aerosol

concentrations is an expression of the importance of the aerosol in driving rCRE.

Using Equations 8 and 9, rCRE can be expressed as a function of LWP and $N_d$. The radiative susceptibility of a cloud to changes in $N_d$ is given by:

$$\frac{drCRE}{dN_d} = \left.\frac{rCRE(1-rCRE)}{3N_d}\right|_{LWP}. \tag{10}$$

Figure 2 shows examples of the theoretical relationships between rCRE and LWP, and between cloud radiative susceptibility and rCRE for different $N_d$: 200 cm$^{-3}$ (blue), 500 cm$^{-3}$ (red), and 1000 cm$^{-3}$ (green). The mean solar zenith angle observed at

the SGP ($\theta_0 = 45°$) was used, and we assumed $g = 0.86$, $T = 300K$ and $p = 1000$ mbar.

Figure 2a shows that for lower LWP values rCRE increases rapidly with increasing LWP. The rate of increase decreases with progressive increase in LWP until the curve begins to saturate. In this example, the saturation begins for rCRE between around 0.7 to 0.8. Complete saturation does not occur at rCRE = 1 due to the diffuse component of the all-sky downwelling shortwave radiation flux. For a very optically thick cloud the direct beam is extinguished but the diffuse component is equal

to the total radiation, assuring that the total radiation transmission does not vanish. Therefore, total radiation extinction does not occur as quickly as might be expected. We also observe a slight increase in rCRE with increasing $N_d$. The rCRE is more



sensitive to changes in $N_d$ at moderate LWP values (between 50 and 100 g m$^{-2}$). Also, for a fixed LWP, the difference between the rCRE obtained for $N_d = 200$ cm$^{-3}$ and $N_d = 500$ cm$^{-3}$ is larger than the rCRE difference obtained using the larger $N_d$ ($N_d = 500$ cm$^{-3}$ and $N_d = 1000$ cm$^{-3}$). The maximum radiative susceptibility occurs at rCRE = 0.5, and is higher for smaller $N_d$ (Fig. 2b). This is consistent with previous results that predict that cleaner clouds are more susceptible to $A_c$

changes than polluted clouds (Platnick and Twomey, 1994). The same authors also report that $A_c$ sensitivity to $N_d$ is a maximum when $A_c$ is 0.5, which is consistent with the larger separation between the curves in the moderate LWP range and for rCRE = 0.5.

### 3.3 Broad statistical analysis of the observations

To understand how the cloud radiative effect responds to changes in different parameters, a broad statistical analysis of the

long-term dataset obtained at SGP was undertaken. As LWP largely dominates rCRE (Eqs. 8 and 9, Fig. 2), the data were binned by rCRE and LWP. The bin sizes were 0.02 for rCRE and 5 g m$^{-2}$ for LWP. For each bin the average of several different variables ($A_i$, $D_i$, $f_c$, LTS, $\tau_c$ and $w'^2$) was calculated. This procedure allows us to isolate the LWP contribution to rCRE and to observe the associations of other properties with rCRE in the third (colored) dimension. To reduce variability due to poor sampling statistics, we require at least 15 points in each 2D-bin. To observe the general trend of rCRE with LWP

and the other variables, for this analysis, the broader LWP range was used.

Figure 3 shows that rCRE presents a clear increasing tendency with LWP, in agreement with the theoretical two-stream approximation, shown in Figure 2. Because the data set is dominated by $f_c \sim 1$, for a fixed LWP, differences in rCRE are primarily due to microphysical influences. Some rCRE differences could be related to the relatively small number of broken cloud events that: i) reduce rCRE due to the smaller $f_c$ associated with this cloud type; and, ii) introduce the possibility of

three-dimensional radiative effects (e.g., Wen et al. 2007), and therefore deviations from the simple two-stream model approximations that form the basis of the rCRE analysis. The distribution of LWP (Fig. 1a) indicates that the number of observations decreases with increasing LWP. Therefore, the larger number of observations at lower LWP results in a larger rCRE spread for low LWP values, compared to the high LWP. This further contributes to the vertical spread of points at low LWP.

For the liquid clouds that meet our analysis criteria, two different cloud types are identified: i) broken-cumulus clouds characterized by lower mean $f_c$ and higher $w'^2$, and ii) stratiform clouds associated with higher $f_c$ and lower $w'^2$. As most broken cumuli are concentrated in the lowest LWP range (usually LWP < 100 g m$^{-2}$) and have lower $f_c$, they generally present smaller rCREs than stratiform clouds (Eq. 4). Since broken cumuli are associated with local convection it is expected that this type of cloud exhibits a higher local coupling with the surface, and therefore a smaller $D_i$, as observed in Figure 3d.

On the other hand, the stratiform clouds at SGP tend to be associated with deeper boundary layers, therefore leading to higher decoupling between the surface and the atmosphere. Stratiform clouds are also controlled by large-scale subsidence and exhibit a higher LTS than broken cumuli (Fig. 3f). The joint probability distribution function of $D_i$ and $f_c$ shows that low $f_c$ cases are only observed when $D_i$ is low (Fig. 4).





Figure 3b shows the strong dependence of $\tau_c$ on LWP, in agreement with Equation (9). The dependence of rCRE on $\tau_c$ is also easily identified. As $\tau_c$ is only retrieved for $f_c > 0.95$, low rCRE values are not observed. For a fixed LWP, rCRE exhibits a weak trend with $A_i$ (Fig. 3a). When LWP is smaller than about 100 g m$^{-2}$, this trend seems to occur in both directions, indicating that both high and low rCRE can be observed in more polluted conditions. One could infer that the positive trend

is due to cloud microphysical changes caused by higher aerosol loading, while the negative trend could be due to the semi-direct effect of aerosol on clouds. However, meteorology also impacts the system and influences the rCRE. For example, different cloud dynamics could be linked to both changes in rCRE and in aerosol concentration. To understand the role that meteorology plays on the rCRE, some dynamical indices are now considered.

Higher turbulence facilitates more efficient droplet activation. Therefore, considering that for a constant LWP, variation in

$A_c$ is due to changes in $N_d$, it is expected that, more turbulence would result in more droplets and higher cloud radiative effect (Feingold et al., 2003). However, Figure 3c shows that for a fixed LWP there is a weak dependence of rCRE on $w'^2$, with higher rCRE usually occurring for *weaker* turbulence. This result suggests that the rCRE is more dependent on macroscale cloud properties such as LWP and $f_c$ than on cloud microphysics. For example, higher turbulence is associated with broken cumuli that present lower $f_c$, and therefore lower rCRE.

The correlation coefficients between the mean $f_c$, LTS and $D_i$ (Fig. 3d-f) were calculated. The correlation between $f_c$ and $D_i$ ($\rho_{fc,Di} = 0.72$) is larger than the correlation between $f_c$ and LTS ($\rho_{fc,LTS} = 0.55$). The correlation between LTS and $D_i$ is also positive, with $\rho_{LTS,Di} = 0.54$. As previously mentioned, LTS and $f_c$ are expected to correlate well for low stratiform clouds. However, as the data in Figure 3 also include some broken clouds, $\rho_{fc,LTS}$ is not as high as in previous assessments that only analyzed stratiform clouds (eg. Klein and Hartmann, 1993, Wood and Bretherton, 2006). We hypothesize that the stronger

$\rho_{fc,Di}$ compared to $\rho_{fc,LTS}$ is a consequence of two factors: (i) $D_i$ is calculated for each cloud element and is therefore closely connected to the local cloud conditions, and (ii) LTS is based on the potential temperature at 700 hPa, which may not always be relevant to the local cloud conditions.

Both meteorological indices used in the analysis, LTS and $D_i$, as well as $f_c$, (Fig. 3d-f) impart a clearer signal in rCRE than does $A_i$ (Fig. 3a). Figures 3d-f show that, on average, the rCRE is larger for less coupled atmospheric conditions, higher LTS

and higher $f_c$, associated with solid stratiform clouds. These results indicate that the cloud radiative effect is more related to macroscopic variables such as LWP and $f_c$ than to changes in aerosol loading and cloud microphysics.

Cloud albedo was also analyzed as a function of LWP and the six other variables analyzed in Figure 3. However, as rCRE is directly proportional to the product of $A_c$ and $f_c$ (Eq. 4) and most of the observations are concentrated at the same cloud fraction bin (Fig. 1b), the results obtained for $A_c$ are very similar to the ones obtained for rCRE and are therefore not shown

here. To isolate the effects of $f_c$ and $A_c$ on rCRE, the variation of $A_c$ with five key variables (LWP, $A_i$, $w'^2$, $D_i$ and LTS) for completely overcast conditions ($f_c = 1$) was analysed (Fig. 5). Figure 5 shows that $D_i$ and LTS have a stronger influence on $A_c$ than does $A_i$. This implies that, besides their association with $f_c$, $D_i$ and LTS also have a direct impact on $A_c$ (and, in turn, on rCRE). On the other hand, for overcast conditions, $A_c$ does not show large variations with $w'^2$ (Fig. 5b). This suggests that





the rCRE trend with $w'^2$ observed in Figure 3c was really associated with the $f_c$ variations observed in the different cloud regimes.

Since high $f_c$ scenes dominate the data (Fig. 1b) and LWP plays a central role in cloud radiative responses, we attempted to identify and compare the signals due to LWP with those due to aerosol on rCRE. Daily correlations between rCRE and these two key variables ($A_i$ and LWP) were analyzed. For this analysis, the LWP range was restricted to avoid drizzle and uncertain retrievals, as explained in section 3.2. Cases that had less than 25 points per day were excluded from this analysis. In the original database, 1093 days fit the low, warm, non-precipitating clouds criteria. After selecting cases that satisfied the minimum requisite number of points per day, and had non-missing coincident retrievals of rCRE, LWP and $A_i$, only 323 days remained. The histograms of the distribution of the correlations between rCRE and $A_i$ ($\rho_{rCRE,Ai}$) and rCRE and LWP ($\rho_{rCRE,LWP}$) are shown in Figure 6.

According to Figure 6a, rCRE and $A_i$ can either be positively or negatively correlated. The proportion of negatively and positively correlated cases is roughly 50%/50% for $\rho_{rCRE,Ai}$. On the other hand, rCRE and LWP show a much higher positive correlation than rCRE and $A_i$ (Fig. 6b). The histograms show that $\rho_{rCRE,Ai}$ is on average $0.00 \pm 0.02$ while $\rho_{rCRE,LWP}$ was on average $0.46 \pm 0.02$. For about 90% of the cases rCRE and LWP are positively correlated. Therefore we can infer that LWP clearly dominates the cloud radiative effect, while the aerosol signal on rCRE is ambiguous. A similar analysis was performed for more decoupled conditions ($D_i \geq 0.5$) and less decoupled conditions ($D_i \leq 0.25$) (Fig. 7). No significant differences were observed for different coupling conditions, supporting the result of Delle Monache et al. (2004) that the relationship between surface aerosol measurements and cloud level aerosol measurements is uncorrelated with the degree of boundary layer vertical mixing at this site.

**3.4 Case studies**

The results shown in the previous sections provide broad insight into the general macroscopic behavior observed for warm clouds at SGP. For a deeper understanding of the processes related to those long-term trends, some cases were further analyzed. Two days that presented relatively high positive or negative correlations between rCRE and $A_i$ were selected and investigated further. The selected case studies have a long time series, with at least 6 hours of rCRE retrievals, in addition to continuous measurements of relevant properties, providing a good sample of observations.

**3.4.1 Case study 1: Positive correlation between rCRE and $A_i$**

Figure 8 shows the time series of several relevant measurements, such as $\tau_c$, LWP, rCRE, $A_i$ and $D_i$, for January 9th 2006. The time series of the vertical profile of radar reflectivity ($Z$) is also shown. Since the rCRE can only be measured during sunlit periods ($\theta_0 < 80°$), this analysis focuses on that period. Due to the detection of multiple layers of clouds after 20 UTC, the plots are restricted to the period from 12 to 20 UTC (6 to 14 LT). The correlation between rCRE and $A_i$ for this day is positive and about 0.75.





The radar reflectivity indicates that this case represents a solid stratiform cloud that begins to develop with the boundary layer at ~12 UTC (Fig. 8b). $h_{CT}$ peaks around 1 km and remains constant after 16 UTC. Note that according to the radar reflectivity it is highly unlikely that this day was affected by precipitation.

The strong positive correlation between rCRE, $\tau_c$ and LWP is also noted (Fig. 8a). As previously pointed out these three variables are closely related (Eqs. 8 and 9). On that day, radiometric measurements were only available after ~14 UTC, so rCRE and $\tau_c$ were only retrieved after that time.

The increase in the incoming solar radiation absorbed by the atmosphere and reaching the surface, warms the atmosphere. The LCL increases with time until it stabilizes at 600 m around 18 UTC. The diurnal cycle of shortwave radiation affects the coupling between the surface and the boundary layer leading to more coupled conditions in the afternoon (Fig. 7d). The relation between $D_i$ and solar radiation is further explored in sections 3.4.2 and 3.5.

After about 16h UTC both $A_i$ and LWP, decrease (Fig. 8a). The mechanisms that lead to the decreases are most likely associated with entrainment and drying as the boundary layer deepens. (The relative humidity time series shows that RH decreases with time, until about 18 UTC, when it stabilizes at about 0.7). Dilution due to the increase in the boundary layer depth likely explains the drop in surface aerosol concentration and decrease in $A_i$.

Next, we aim to understand how the co-variability between LWP and $A_i$, could be linked to the response of rCRE to these two variables. Figure 9a-c shows the correlations between rCRE and $A_i$ ($\rho_{rCRE,Ai}$), rCRE and LWP ($\rho_{rCRE,LWP}$) and LWP and $A_i$ ($\rho_{LWP,Ai}$) for the selected day. Only points that have coincident measurements of all three variables – rCRE, LWP and $A_i$ – are used. The number of valid points is 329.

For this day, all correlations are positive, with $\rho_{rCRE,Ai} = 0.75$, $\rho_{rCRE,LWP} = 0.82$ and $\rho_{LWP,Ai} = 0.50$. The results and theory shown in sections 3.2 and 3.3, indicate that the changes in LWP drive changes in rCRE. However, microphysical responses also need to be considered. For a vertically homogeneous cloud, $r_e$ can be calculated as a function of LWP and the $\tau_c$ (Stephens, 1978).

$$r_e = 1.5 \frac{LWP}{\tau_c}, \qquad (11)$$

where LWP is given in g m$^{-2}$ and $r_e$ is given in μm.

For a cloud with constant LWP, a measure of the strength of aerosol-cloud interaction ($\alpha$) can be obtained from the relative change between droplet effective radius ($r_e$) and $A_i$:

$$\alpha = -\left. \frac{\partial ln r_e}{\partial ln A_i} \right|_{LWP}. \qquad (12)$$

According to this definition, $\alpha$ is expected to be positive and vary between 0 and 0.33, with a typical value of 0.23 (Feingold et al., 2001).

To assess the microphysical effect of aerosols on clouds, $r_e$ was calculated using Equation 11 and plotted as a function of $A_i$. In an attempt to isolate the aerosol effects on $r_e$, the dataset was divided into three LWP bins. For each bin, the linear





regression between the logarithm of $r_e$ and logarithm of $A_i$, was obtained. The slope of each linear fit provides the parameter $\alpha$ (Fig. 9d).

For this case, $r_e$ varied between 2 and 7 μm and $\alpha$ is positive, as expected. The values obtained for $\alpha$ are within the expected range, except for the higher LWP category (Fig. 9d). However, there is a large variability in the magnitude of $\alpha$. For the

highest LWP range, $\alpha$ is about twice the value obtained for the mid-range LWP.

The question remains whether the positive correlation between rCRE and $A_i$ is a result of the positive correlation between rCRE and LWP observed on that and many days in this data set (Fig. 3) – i.e., a macrophysical response – or whether it is due to the negative correlation between $r_e$ and $A_i$ – i.e. a microphysical response. This single case study suggests that both contributions are possible, but raises concerns about being too reliant on the microphysical response as an indicator of

aerosol-related rCRE.

### 3.4.2 Case study 2: Negative correlation between rCRE and $A_i$

A case that shows a high negative correlation between rCRE and $A_i$, April 26[th] 2006, was also selected and analyzed in detail. Similar to the previous case, Figure 10 shows the time series of some of the relevant measurements and retrievals for this day. As the cloud completely vanished during late afternoon the analysis timeframe was once again restricted to between

12 and 20 UTC. The radar profile is shown from earlier in the day (5 UTC and on), as some drizzle was detected during nighttime. The drizzle may have scavenged the aerosol particles and could explain the low $A_i$ values shown in Figure 9c, through ~1450 UTC. The red line indicates daytime in Figure 10b.

Once again, a strong positive correlation between rCRE, $\tau_c$ and LWP is observed.

The evolution of $D_i$ is similar to the previous case, indicating that for both days the coupling between atmosphere and

surface is driven by the diurnal cycle of radiation, rather than by other variables. This day was much warmer than the previous case and presented higher LCL values and lower surface RH. The surface temperature differences between the two days varied from 6 K to 10 K during the period analyzed.

The temporal evolution of LWP and the vertical profile of reflectivity for April-26-2006 (Fig. 10b-c) indicate that at about 14 UTC the stratiform cloud begins to dissipate, transitioning to broken cumuli after ~17 UTC. The decrease in both LWP

and $f_c$ after 14h UTC coincides with an increase in $A_i$. One hypothesis to explain this behavior is that boundary layer deepening and entrainment drying reduce cloud amount as the day progresses. $D_i$ decreases because when clouds do form (a prerequisite for calculating $D_i$) the local coupling is relatively strong. The increase in $A_i$ from a low post-drizzle clean atmosphere could be a result of a combination of surface sources, transport, and entrainment of free tropospheric air. It is also possible that cloud breakup may be caused by the aerosol semi-direct effect, however $A_i$ was lower on this day and the

analysis of the Ångström exponent and single scattering albedo (SSA) indicate that there are no significant differences in aerosol intensive properties (and thus, perhaps in aerosol type) between this and the previous case. The mean Ångström exponent at 1 μm cut size for case 2 was $2.274 \pm 0.010$, while in the previous case it was $2.107 \pm 0.008$. The mean SSA was





0.9721 $\pm$ 0.0012 and 0.9826 $\pm$ 0.0004, for case 2 and case 1, respectively. The difference in the uncertainty indicates that for case 2, both the Ångström exponent and SSA fluctuate more. Finally, while one might want to invoke a role for the increasing aerosol evaporating smaller droplets more efficiently, which in turn would decrease $f_c$ (Small et al., 2009), these aerosol loadings are relatively low, and as already discussed in section 3.3, many other dynamical features influence $f_c$ and

cloud development, especially during the daytime.

The correlations between rCRE, LWP and $A_i$ for case 2 are shown in Figure 11a-c. The microphysical effect of aerosol on drop size is shown in Figure 11d. The number of valid points for this study case is 204.

The correlation between rCRE and $A_i$ is negative and equal to -0.65 for this case. The correlation between rCRE and LWP is 0.64, smaller than in the previous case study, but still positive, as expected. Figure 11c shows that for case 2, LWP and $A_i$ are

negatively correlated with $\rho_{LWP,Ai}$ = -0.44.

The $r_e$ retrievals indicate that the sizes of most of the droplets analyzed in this case fall in the same range as the previous case study (between 3 and 10 μm). Here, however, $\alpha$ is negative (Fig. 11d), for which there is no physical explanation given the stratification by LWP and our expectation that drop size decreases with an increasing number of CCN for the same amount of condensed water (Twomey, 1977). This unexpected behavior could derive from a combination of factors: uncertainty in

measurements, uncertainty in linear fits, and possibly the rather broad LWP binning, among others.  Given the unphysical $r_e$ response to increasing aerosol, the positive correlation between rCRE and LWP, and the overwhelming contribution of macroscopic and dynamical variables to the cloud system compared to the aerosol signal discussed in section 3.3, the results indicate that the observed negative correlation between rCRE and $A_i$ is most likely due to the fact that LWP and aerosol are negatively correlated, presumably due to independent factors.

Most techniques employed to retrieve $\tau_c$ using ground-based instruments rely on overcast conditions (eg., Barnard et al., 2008, Min and Harrison, 1996). Xie and Liu's (2013) technique can be used to retrieve $\tau_c$ for lower cloud coverage. In Figures 9d and 11d, $r_e$ was calculated using retrievals of $\tau_c$ from a broadband radiometer (RFA) following Barnard and Long (2004). Additionally, two other methods were used to retrieve $\tau_c$ and $r_e$ for the case studies highlighted above: the Multi-Filter Rotating Shadowband Radiometer (MFRSR, Turner and Min, 2004) and broadband radiometer retrievals by Xie and

Liu (2013). Effective radii $r_e$, determined from the measured LWP and each of the $\tau_c$ retrievals, were used to obtain the aerosol-cloud interaction ($\alpha$) slope (Table 2). Retrievals acquired when $\theta_0$ > 70° were excluded from this analysis as the measurements are less reliable at higher solar zenith angles and the retrievals diverged greatly at high $\theta_0$ in some cases. The different methodologies used to retrieve $\tau_c$ result in different $\alpha$, and, for some cases, even the sign of the slopes disagree. The difference observed for $\alpha_{\square\square}$ estimates shown in Table 2 compared to Figures 9 and 11, is due to the restriction of co-

location of data points among the three datasets and the $\theta_0$ < 70° threshold.

As emphasized above, this comparison raises concerns about reliance on $\alpha$ to quantify aerosol-related rCRE in terms of microphysical metrics. The requirement of binning by LWP leaves low statistics for calculating slopes in each bin and uncertainties in the slopes are high. Given the low statistics, differences in the retrievals can result in the large differences in





$\alpha$ seen here, including changes in sign. These microphysical measures are useful for detecting aerosol effects on cloud properties, but are best used in conjunction with other measurements to fully understand the relevant physical processes. Using these measures for quantification of the aerosol indirect effect (the aerosol induced cloud radiative effect), especially in case studies where statistics are low, can be misleading. Studies that provide larger statistics may produce more

meaningful quantifications (e.g., McComiskey et al. 2009), but will still contain biases inherent in any retrievals used to provide input properties to the calculation.

### 3.5 Further generalizations

The diurnal cycles of the $D_i$, shown in two case studies of section 3.4, were very similar, with higher $D_i$ in the morning and lower $D_i$ around 20 UTC (Figs. 8d and 10d). To verify if this trend is generally observed, the complete time series obtained

during this 14-year study was used. The dataset was divided into 0.5-hour bins and the mean diurnal cycle of $D_i$ during daytime was analyzed (Fig. 12).

Figure 12 shows that the temporal evolution of $D_i$ is strongly linked to the diurnal cycle of solar radiation. On average, the atmosphere is highly decoupled in the morning. As the sun rises, the surface gets warmer, and solar energy is transferred from the surface to the atmosphere, favoring more coupled conditions (lower $D_i$). The higher coupling between the surface

and the atmosphere increases turbulence. As the incoming solar radiation during the afternoon decreases, the atmosphere gradually cools. After ~ 20 UTC, the boundary layer collapses leading to less coupled conditions in the late afternoon.

The results shown in the previous section also indicate that, for these two case studies, the correlation between rCRE and $A_i$ has the same sign as the correlation between LWP and $A_i$ (Figs. 9 and 11). For the first case study, $\rho_{rCRE,Ai}$ and $\rho_{LWP,Ai}$ is positive, while for the second case study both correlations are negative. This suggests that the sign of $\rho_{rCRE,Ai}$ is mainly

determined by $\rho_{LWP,Ai}$. We now test the validity of this hypothesis and if this statement can be expanded for the whole dataset. For each day the correlation between rCRE and $A_i$ ($\rho_{rCRE,Ai}$) and between LWP and $A_i$ ($\rho_{LWP,Ai}$) were calculated. Figure 13 shows the results obtained for these correlations, where each point represents one day. This was done for the 323 days that had coincident measurements of the three variables ($A_i$, LWP, and rCRE). An orthogonal linear fit of the observations was performed.

Figure 13 shows that this statement can be generalized. Usually, if $A_i$ and LWP are positively (negatively) correlated, the correlation between rCRE and $A_i$ is positive (negative). This relationship was further analyzed as a function of several variables ($A_i$, LWP, $D_i$, $\tau_c$, wind direction, wind speed, surface RH, $w'^2$), none of which significantly influenced the results. Considering all the days analyzed, the correlation between $\rho_{rCRE,Ai}$ and $\rho_{LWP,Ai}$ is 0.54. This result suggests that the aerosol signal observed in rCRE based on daily correlations may often be a misinterpretation of the positive relationship between

rCRE and LWP. Once again, for the data set analyzed, which consists overwhelmingly of high $f_c$ events, the cloud radiative effect appears to be predominantly driven by macroscopic variables rather by microphysical responses.



Given the uncertainty in calculations of $\alpha$ (Table2) the current work sounds a cautionary note regarding placing too much emphasis on microphysical metrics. This does not exclude the possibility of an aerosol influence on the cloud radiative effect but suggests that careful analysis should be done to quantify macrophysical relationships, such as those shown here. Moreover, consideration of the co-variability in aerosol and meteorological conditions has a strong influence on the

detectability of aerosol-induced rCRE and therefore deserves attention (George and Wood 2010; Feingold et al. 2016).

## 4 Summary and conclusions

A comprehensive study was performed to understand the relative effects of aerosols and meteorological drivers on the radiative effect of low-level clouds. Fourteen years of coincident ground-based clouds, aerosol and meteorological measurements over the SGP were analyzed. The impact of different physical properties on the instantaneous cloud radiative

effect was studied. The dataset was divided into rCRE and LWP bins and the mean values of properties such as $f_c$, $\tau_c$, $D_i$, LTS, $A_i$ and turbulence were analyzed. Most of the data are characterized by high $f_c$ so that rCRE is predominantly a function of $A_c$ (Eq. 4), which is in turn a strong function of LWP, and to a lesser extent drop concentration (Eqs. 7 and 9). Whereas a strong dependence of rCRE on LWP is clearly identified, the average over the whole dataset shows a weak influence of aerosol on rCRE. For low LWP, polluted conditions are associated with both high and low rCRE. The impact of LTS, and $D_i$

on rCRE is also stronger than the impact due to aerosol particles.

Since LWP is such a key driver of rCRE, the impact of aerosols and LWP on the cloud radiative effect were compared by assessing the daily correlations between rCRE and $A_i$ and rCRE and LWP. While the daily distribution of $\rho_{CRE,LWP}$ shows a clear positive signal, the daily distribution of $\rho_{rCRE,Ai}$ is centred around 0, confirming the previous statement that high aerosol concentrations can be associated with both higher and lower rCRE.

Case studies that showed both positive and negative correlations between rCRE and $A_i$ were further investigated. For these two selected days, rCRE was positively (negatively) correlated with $A_i$ when $A_i$ and LWP were positively (negatively) correlated. This behavior can be generalized to the other days analyzed. The case studies also show that microphysical metrics to estimate aerosol-cloud interaction (Eq. 10) are very uncertain and reliance on these estimates to quantify aerosol-related rCRE can be misleading.

The diurnal cycle of $D_i$ over the SGP is strongly driven by the diurnal cycle of solar radiation. Both, LTS and $D_i$ are highly correlated with $f_c$ however $\rho_{fc,Di}$ is larger than $\rho_{fc,LTS}$. This is because LTS and $f_c$ are tightly related for stratiform cloud, but less so for broken clouds. On the other hand, $D_i$ represents both cloud types well because it is calculated for individual cloud elements. Stratiform clouds are usually observed early in the morning, when the boundary layer is less coupled due to the smaller sensible heat flux. As the surface warms up, turbulence and therefore surface-atmosphere coupling increases, and

broken cumuli that have smaller $f_c$ are formed.

The results presented here indicate that to first order, macroscopic variables such as cloud condensate and $f_c$ rather than cloud microphysics are the properties that determine the cloud radiative effect. Clearly the aerosol can play a role by modifying




drop size and influencing how LWP manifests in $\tau_c$ and $A_c$. However, while LWP and $f_c$ present a clear signature on rCRE, the aerosol signal is barely distinguishable. The aerosol signal is also difficult to quantify because of the uncertainty in calculation of the metrics derived from different methods (Table 2, Figs. 9d and 11d) and platforms (McComiskey and Feingold 2012). Future studies that focus on understanding the role of dynamics and other meteorological drivers that

potentially alter the macroscopic cloud properties will be reported on in the near future.

*Acknowledgements*

The authors would like to thank the ARM (Atmospheric Radiation Measurement) Program for processing and providing the data sets used in this work. This work was supported by FAPESP grants 2014/04181-2 and 2013/08582-9, the U.S. Department of Energy's Atmospheric System Research (ASR) program by Grant DE-SC0014568 and by NOAA.

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



**Table 1: List of the measurements, retrievals and ARM instruments at the Southern Great Plains used in this study.**

| Instrument | Resolution in the original data set | Measurement / Retrieval |
|---|---|---|
| Milimeter Wavelength Cloud Radar (MMCR) | 10 s | Column Maximum Reflectivity ($Z_{max}$) |
| Ceilometer / Micropulse Lidar (MPL) | 10 s | Cloud base height ($h_{CB}$) |
| MMCR / MPL | 10 s | Cloud top height ($h_{CT}$) |
| MMCR + Ceilometer | 10 s | Doppler vertical velocity at $h_{CB}$ ($w$) |
| Microwave Radiometer (MWR) | 20 s | Liquid water path (LWP) |
| Broadband radiometers | 1 min | Relative cloud radiative effect (rCRE) <br> Cloud optical depth ($\tau_c$) <br> Cloud fraction ($f_c$) <br> Cloud albedo ($A_c$) |
| Nephelometer | 1 min | Scattering at 550 nm ($\sigma_{550nm}$) <br> Ångström exponent ($Å$) |
| Meteorological station (MET) | 1 min | Lifting condensation level (LCL) |
| Radiosondes + MET + MWR + Models | 1 min | Lower tropospheric stability (LTS) |





**Table 2: Slopes $\alpha$ and their uncertainty obtained using different $\tau_c$ retrievals: from the Radiative Flux Analysis (RFA) , using the Xie and Liu technique (2013, XL) and using MFRSR measurements. Coincident retrievals of $\tau_c$ from each retrieval acquired when $\theta_0 < 70º$, for each day were used to calculate $\alpha$.**

|  | $LWP\ (g\ m^{-2})$ | $\alpha_{RFA}$ | $\alpha_{XL}$ | $\alpha_{MFRSR}$ |
|---|---|---|---|---|
|  | 50 - 75 | $0.27 \pm 0.09$ | $0.32 \pm 0.09$ | $0.23 \pm 0.07$ |
| *Case study 1* | 75 - 100 | $0.26 \pm 0.07$ | $-0.03 \pm 0.08$ | $0.25 \pm 0.06$ |
|  | 100 - 150 | $0.73 \pm 0.26$ | $0.58 \pm 0.30$ | $0.70 \pm 0.24$ |
|  | 50 - 75 | $-0.01 \pm 0.09$ | $0.31 \pm 0.07$ | $0.10 \pm 0.06$ |
| *Case study 2* | 75 - 100 | $-0.09 \pm 0.04$ | $0.25 \pm 0.04$ | $0.07 \pm 0.03$ |
|  | 100 - 150 | $-0.23 \pm 0.04$ | $0.11 \pm 0.02$ | $-0.03 \pm 0.02$ |





**Figure 1: Statistical distributions of: a) liquid water path (LWP), b) cloud fraction ($f_c$), c) rCRE, d) cloud albedo ($A_c$), e) cloud optical depth ($\tau_c$), f) cloud thickness, g) cloud base height ($h_{CB}$), h) cloud top height ($h_{CT}$), i) aerosol index ($A_i$), j) $w'^2 = [w-w_0]^2$, k) decoupling index ($D_i$), l) lower tropospheric stability (LTS).**







**Figure 1: continued.**



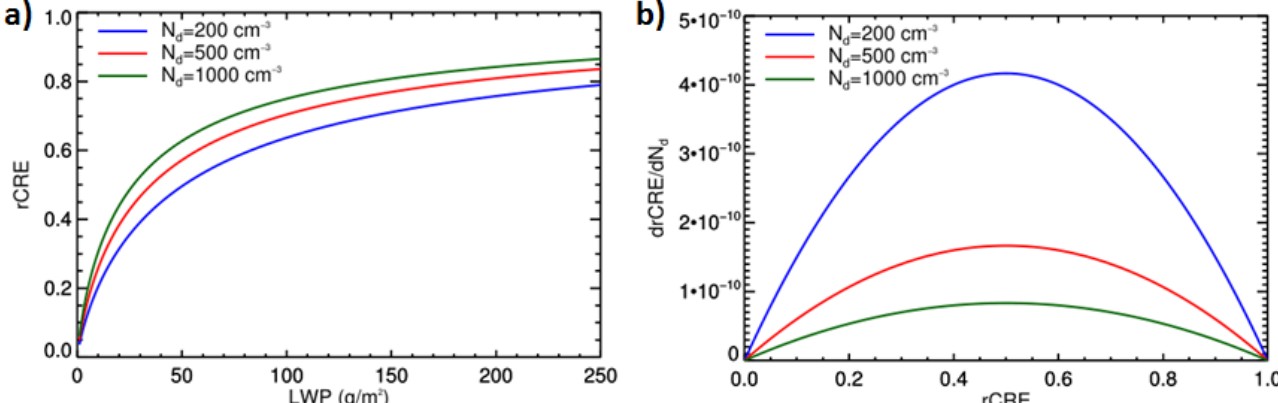

**Figure 2: Theoretical approximations of a) rCRE as a function of LWP, and b) cloud radiative susceptibility to $N_d$ as a function of rCRE for different droplet concentrations: $N_d = 200$ cm$^{-3}$ (blue), $N_d = 500$ cm$^{-3}$ (red) and $N_d = 1000$ cm$^{-3}$ (green).**





**Figure 3: Relative cloud radiative effect as a function of liquid water path colored by a) aerosol index, b) cloud optical depth, c) $w'^2$, d) decoupling index, e) cloud fraction and f) lower tropospheric stability.**



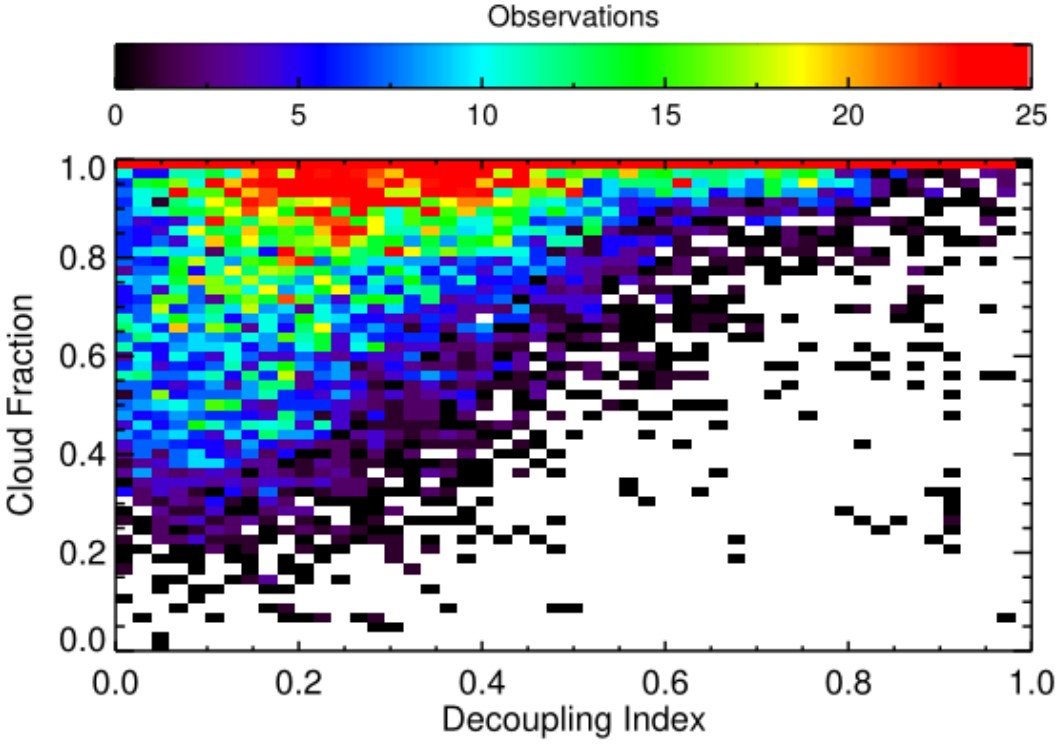

**Figure 4: Joint probability distribution function of $D_i$ and $f_c$ obtained from 14-years of observations at SGP.**





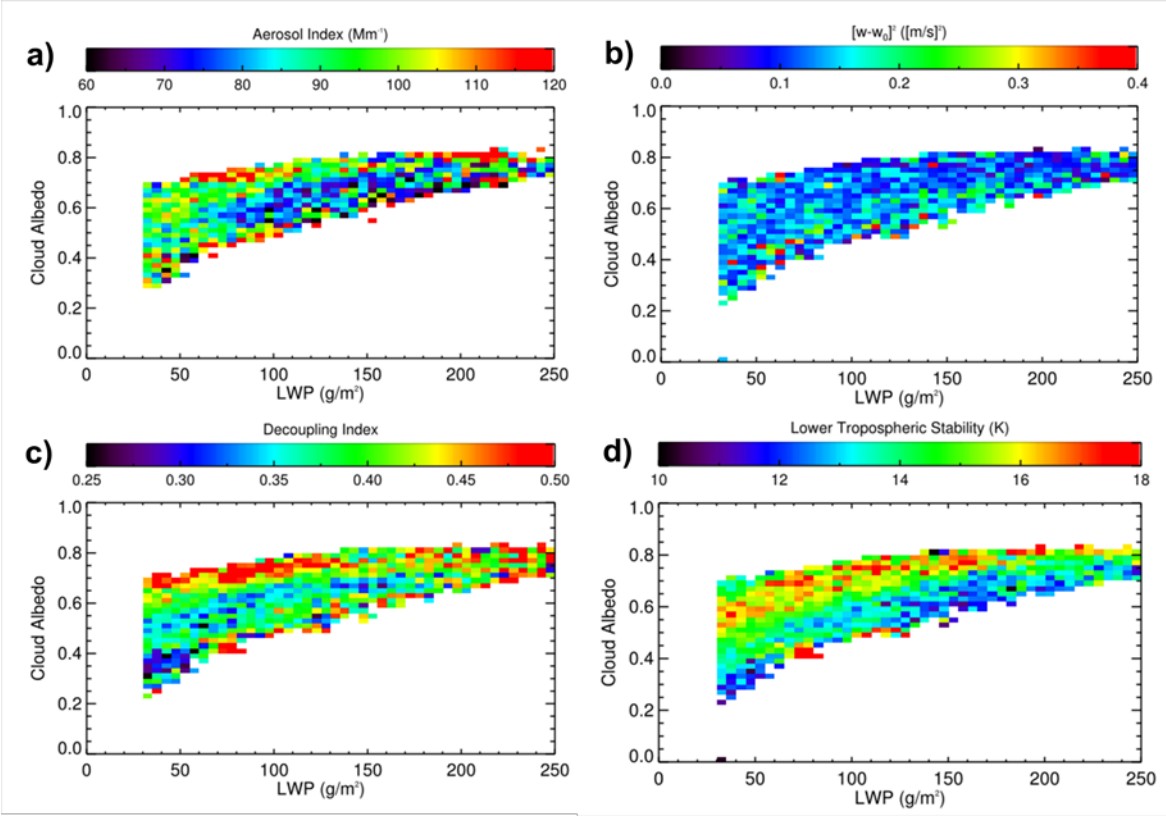

**Figure 5: Cloud albedo as a function of liquid water path colored by a) aerosol index, b) $w'^2$, c) decoupling index and d) lower tropospheric stability, for completely overcast conditions ($f_c = 1$).**





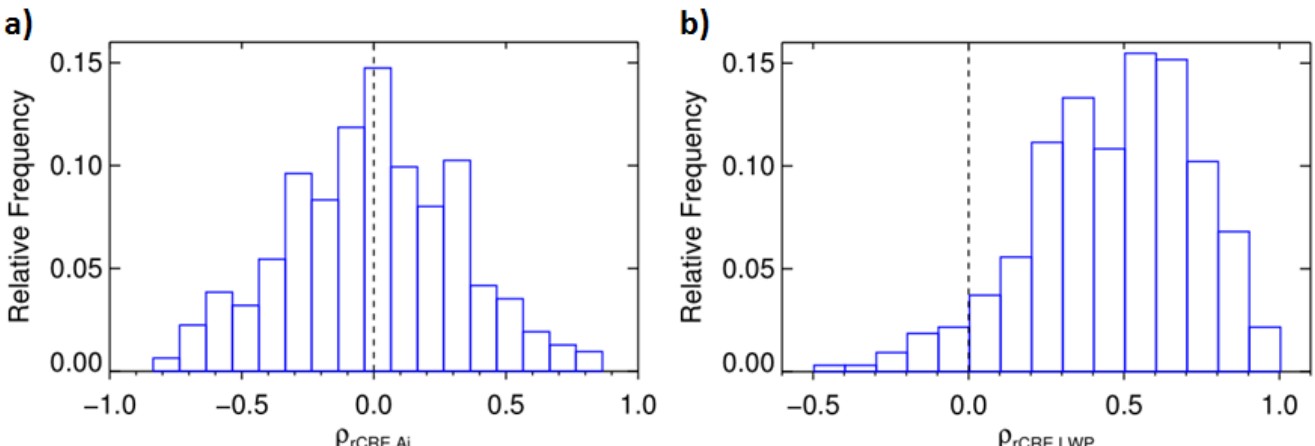

**Figure 6: Daily distribution of the a) correlation between the relative cloud radiative effect (rCRE) and aerosol index ($A_i$) and b) the correlation between rCRE and liquid water path (LWP).**





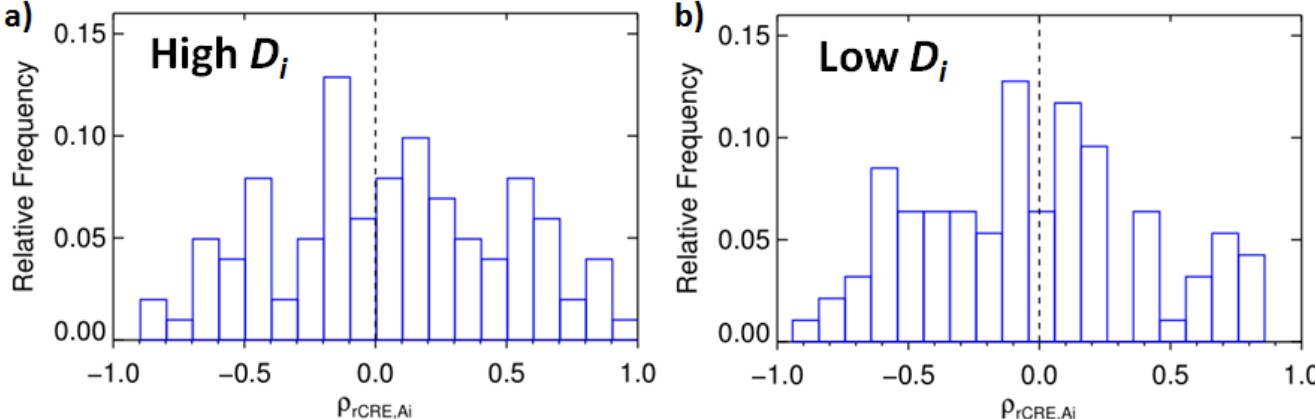

**Figure 7: Daily distribution of the a) correlation between the relative cloud radiative effect (rCRE) and aerosol index ($A_i$) for a) higher decoupling index: $D_i \geq 0.5$, and b) lower decoupling index: $D_i \leq 0.25$.**





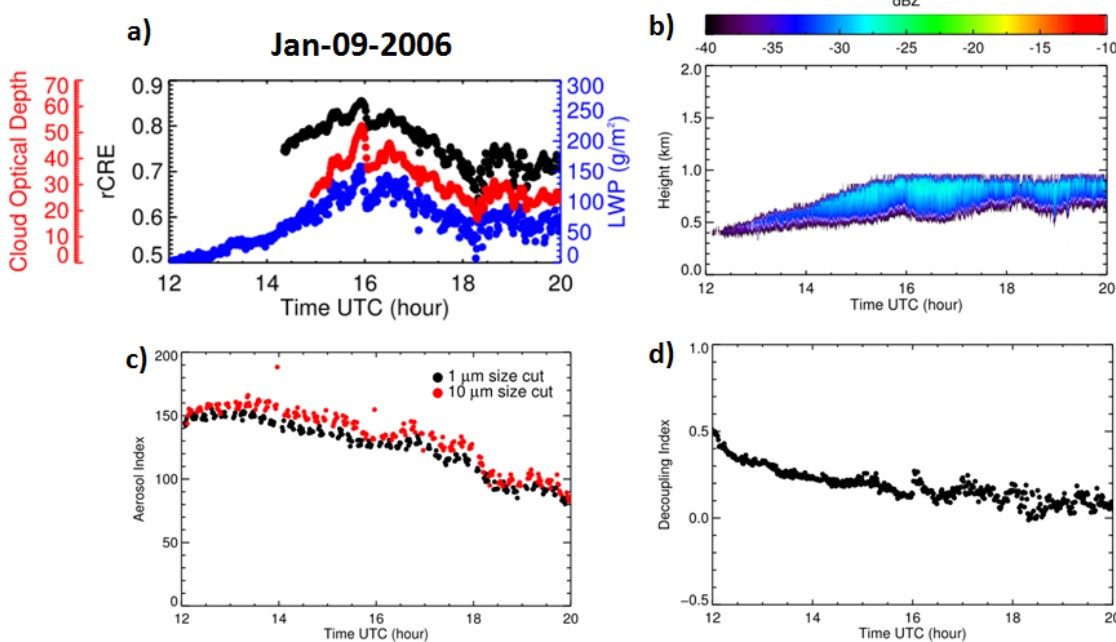

**Figure 8: Time series of: a) rCRE, cloud optical depth and LWP, b) vertical profile of radar reflectivity, c) aerosol index, and d) decoupling index for January 9th 2006.**



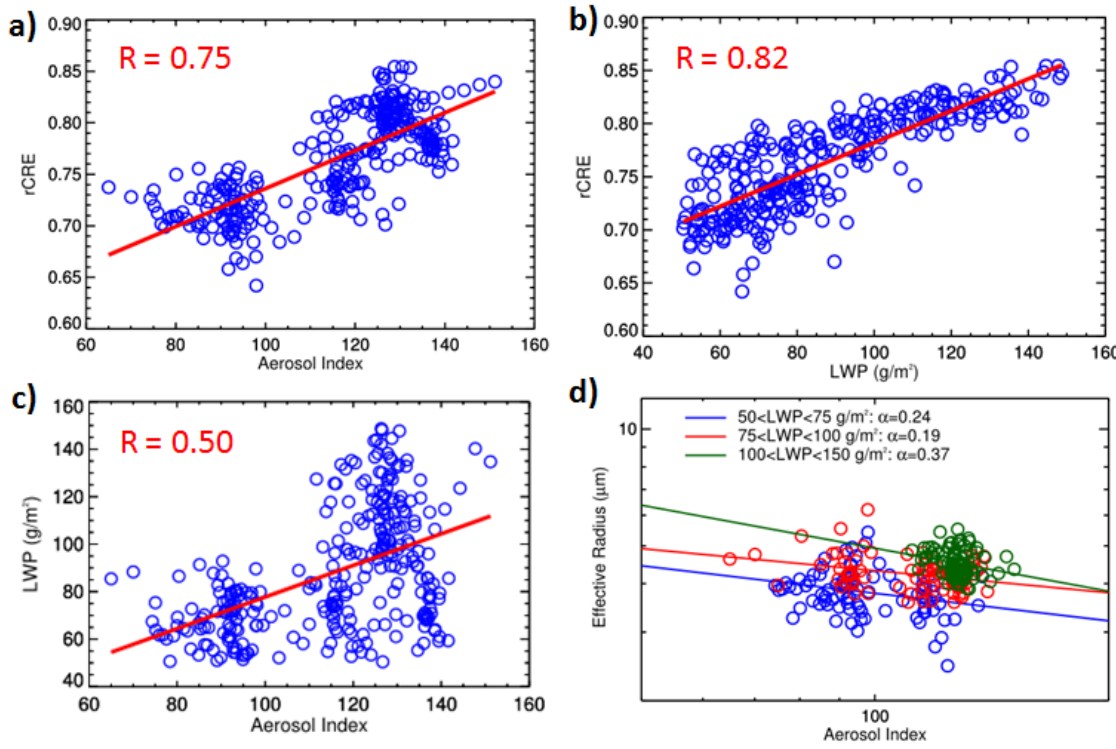

**Figure 9: Correlation between a) rCRE and $A_i$, b) rCRE and LWP, c) LWP and $A_i$ and d) effective radius as a function of $A_i$ grouped by LWP for January 9th 2006.**





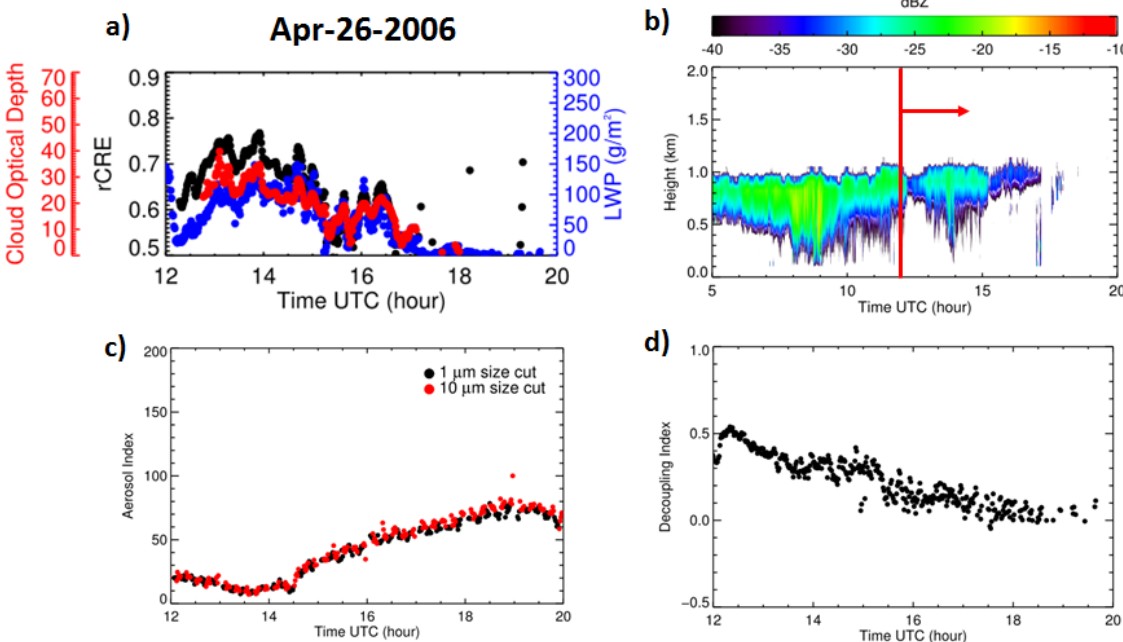

Figure 10: Time series of: a) rCRE, cloud optical depth and LWP, b) radar reflectivity, c) aerosol index, and d) decoupling index for April 26[th] 2006.





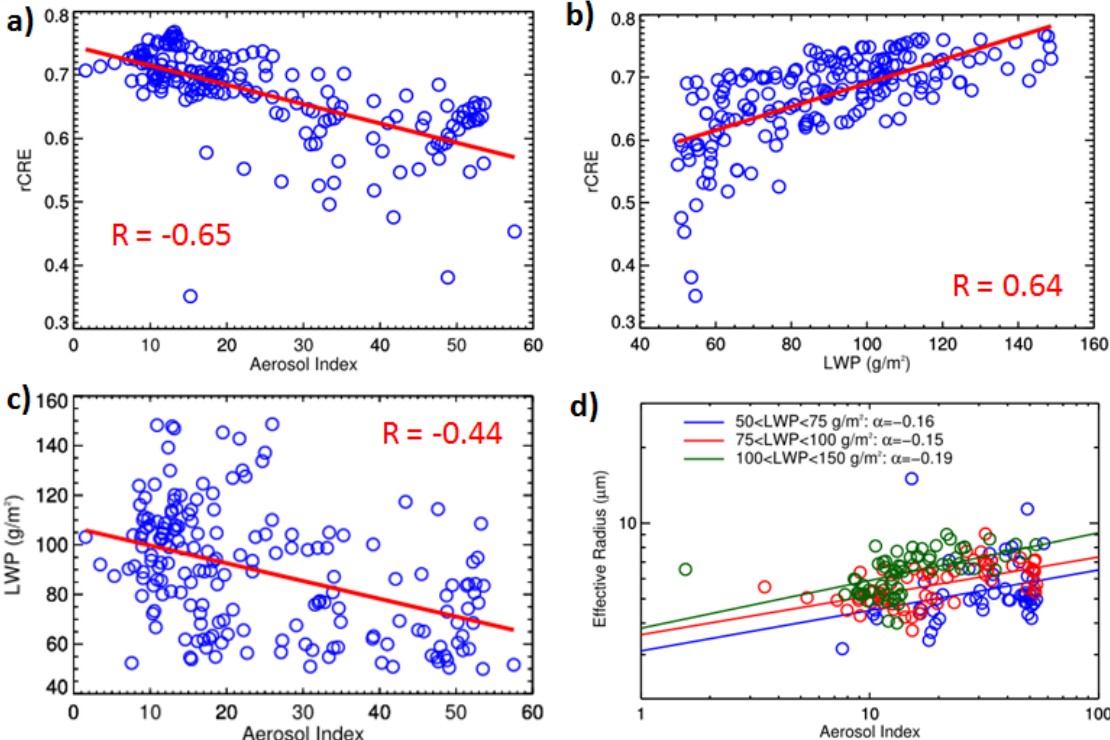

**Figure 11: Correlation between a) rCRE and $A_i$, b) rCRE and LWP, c) LWP and $A_i$ and d) effective radius as a function of $A_i$ grouped by LWP for April 26th 2006.**





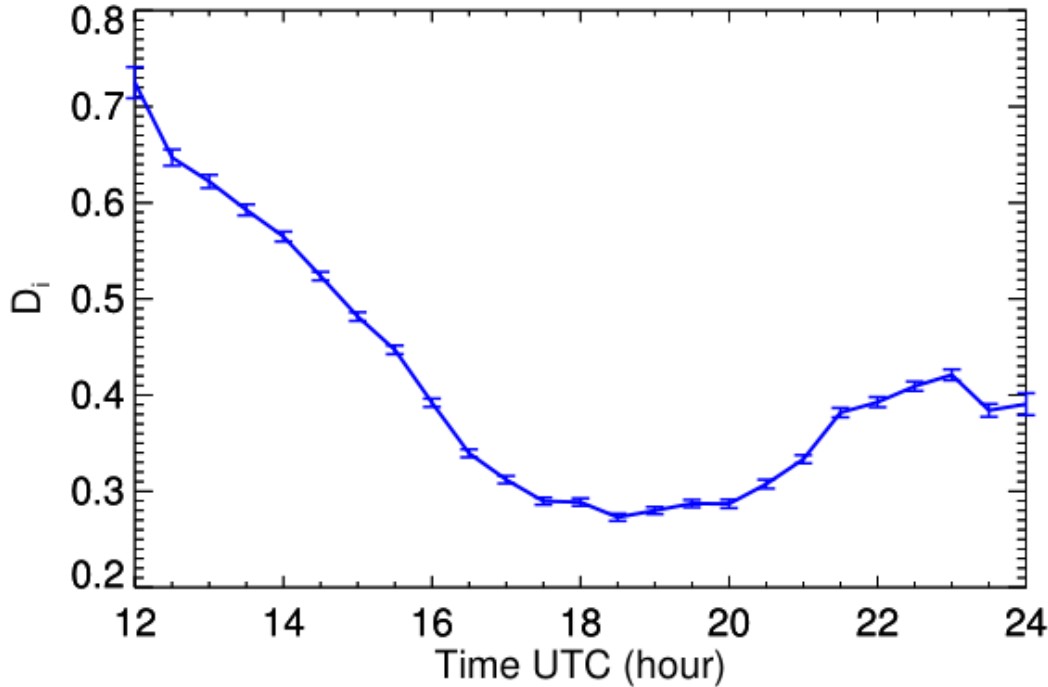

**Figure 12: Mean diurnal cycle of the decoupling index ($D_i$) obtained using 14 years of retrievals at the SGP. Error bars indicate the standard deviation of the mean for each time bin.**





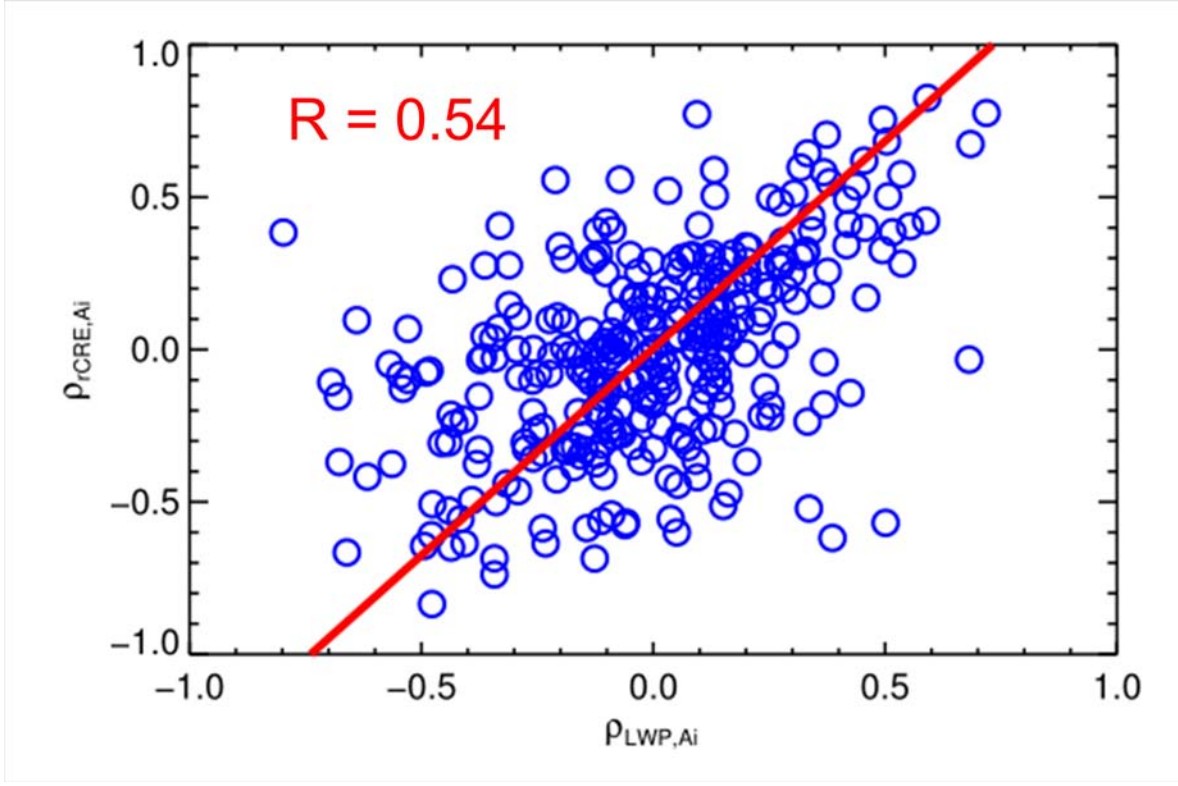

**Figure 13: Correlation between rCRE and $A_i$ ($\rho_{rCRE,Ai}$) versus the correlation between LWP and $A_i$ ($\rho_{LWP,Ai}$).**