# Peer review of "A long-term study of aerosol-cloud interactions and their radiative effect at the Southern Great Plains using ground-based measurements"

_Atmospheric Chemistry and Physics, 2016_

## Referee Comment (RC1) · Anonymous Referee #1 · 7 May 2016

Overall summary:

This manuscript studys how the cloud radiative effect responds to changes in aerosols (aerosol index) and meteorological parameters, including cloud fraction fc, cloud optical depth $\tau$ , decoupling index Di, lower tropospheric stability LTS, and turbulence w'2 by using long-term ground-based measurements from the Atmospheric Radiation Measurement (ARM) Program over the Southern Great Plains. Herein, the cloud type was constrained to shallow liquid water clouds. This work presents valuable information that the impact of macroscopic variable on aerosol-cloud interaction is stronger than the impact due to aerosol particles. Some minor questions/suggestions need to be solved are listed in the following:

[Figure]

Comment and Question:

1. Page 2, Line 4: Absorbing aerosol could also modify the atmospheric temperature profile and stability, and reduce cloud amount via the semi-direct effect (e.g., Koren et al., 2008). Koren et al. (2008) just provide the cloud amount change with aerosol optical depth but didn't show absorbing aerosol could modify the atmospheric stability. Huang et al. (2009) use the Fu-Liou radiation model and CERES radiation flux to derived the heating rate of aerosol layer and directly show the changes in temperature profile.

Huang J., Q. Fu, J. Su, Q. Tang, P. Minnis, Y. Hu, Y. Yi, and Q. Zhao, 2009: Taklimakan dust aerosol radiative heating derived from CALIPSO observations using the Fu-Liou radiation model with CERES constraints,Atmos. Chem. Phys., 9, 4011-4021.

2. Authors used the aerosol index instead of CCN concentration in this study. As the SGP site do equipped with the Cloud Condensation Nuclei Particle Counter, why do not use this data directly?

3. Authors mentioned that all of the relevant variables were averaged to 1-minute resolution. Does this time resolution is suitable for this study?

4. Koren et al. (2008, ACP) show $\omega$ (550 hPa) and RH(350 hPa) yielded the highest correlations with the satellite-derived cloud properties, these parameters will be used to represent the primary meteorological controls on the cloud system. Herein, authors examined the impact of cloud macroscopic properties (fc and $\tau$) and meteorological variables (Di, LTS, and w'2) on cloud radiative effect. How did authors choose those variables?

---

## Referee Comment (RC2) · J. Quaas (Referee) · 25 May 2016

Sena et al present a study relating temporal variability in cloud radiative effects to variability in various quantities including near-surface aerosol index, microwave-retrieved cloud liquid water path, as well as boundary-layer decoupling index and lower-tropospheric stability from 1-min resolution data with a conditioned sampling of low-level non-precipitating liquid-water clouds from 14 years of data at the ARM SGP site. The central result, presented in Fig. 6a, is that the surface aerosol index is uncorrelated with the cloud radiative effect, indicating a negligible aerosol influence on cloud radiative effect. This comes along with Fig. 6b demonstrating the dominant effect of LWP variability on the cloud radiative effect. The result is corroborated by

[Figure]

Fig. 13 showing that sensitivity of the cloud radiative effect to aerosol variability is driven by sensitivity of LWP to aerosol variability.

Very usefully, the authors proceed with a more detailed analysis. Eq. 8 and 9 set the theoretical framework of the discussion, namely the fact that the cloud radiative effect, rCRE, is influenced by three state variables: cloud fraction, LWP, and cloud droplet number concentration, $N_d$. The theoretical considerations (as well as experience and previous literature) clearly show that cloud fraction is the dominant influence, with LWP of slightly lesser but still very large importance, and that variability in $N_d$ has a rather weak influence.

No measurements in $N_d$ are used in the study by Sena et al.. Rather, near-surface nephelometer measurements of the aerosol index are used as a proxy for cloud condensation nuclei concentrations (CCN) and subsequently for $N_d$. The results of the decomposition analysis to identify impacts of the various parameters on rCRE are presented in Fig. 3 where all three effects are convolved. More instructive still is Fig. 5 where only overcast cases are selected. At the theoretical level (Eq. 8 and 9), for a given LWP bin, rCRE can only be a (strictly monotonically increasing) function of $N_d$. In light of this, the results are puzzling. There is no clear relationship of rCRE at given LWP with near-surface aerosol index (Fig. 5a). There is no influence of $w'$ on rCRE (Fig. 5b). There is, however, a more systematic influence of the decoupling index and also of lower-tropospheric stability (LTS). The authors interpret that the aerosol impact is small. This is a straightforward interpretation of Figs. 5a and 6a. But how can this conclusion be true? Does this not imply the simple theoretical model in Eq. 8 and 9 is wrong? The other possible explanation is certainly that the nephelometer measurements near the surface are not a good proxy for in-cloud $N_d$. Perhaps one could test this by trying to relate remote sensing retrievals of $N_d$ to the aerosol index?
[Figure]

A very interesting question is further, why is there the impact of decoupling index (Fig. 5c) and LTS (Fig. 5d), but not of $w'$?

My own experience would point to spurious variability in LWP and cloud fraction that the binning into bins of 5 g m$^{-2}$ and the constraint of the retrieved cloud fraction at 100% is not able to completely inhibit. Given that despite the length of the data record not overly much data is available due to the conditioned sampling, an option would be to use coarser bins and see whether the effects are larger.

Else it is possible that the retrievals of either rCRE or LWP somehow depend on decoupling index or on LTS, but I am not enough of an expert on the retrievals to say whether this is possible. Another possibility is that the diurnal cycle in the decoupling index (Fig. 12) impacts rCRE more strongly than one might anticipate via the not-eliminated impact of the solar zenith angle (Eq. 8).
par

It would be very useful if the authors could discuss these questions in order to understand to which extent the "negative" or "null" result of no influence of aerosols on rCRE is a robust finding.

The authors also conclude that microphysical metrics in general are misleading. This conclusion mainly stems from the results in Table 2. While there is probably consensus that retrievals of cloud microphysical quantities are error-prone, such a broad conclusion should be corroborated better. If the authors decide this is a focus of the paper, then they should include a much more detailed discussion and ideally evaluation of the three different retrieval algorithms for the effective radius and explain why all three are (equally?) valid. In this regard, it should be noted that retrievals of microphysical quantities from the surface in general are more difficult

than from the top of the atmosphere (Brueckner et al., J. Geophys. Res., 2014 doi: 10.1002/2014JD021775)

Besides these two main comments, I have a few specific comments listed below.

In general I think this paper is a very useful contribution to the important discussion on how statistical analysis of remote sensing data may be used to quantify aerosol-cloud interactions. The paper is very well written and the figures are excellently prepared. The topic is highly pertinent to Atmos. Chem. Phys.

**Specific comments**

Title: Why not name the ARM SGP site?

Abstract p1 l18: I think that in the abstract "weak" needs quantification. One would expect the aerosol to be second order anyway.

P1 l29: Most publications currently would suggest additional effects due to microphysical adjustments, not so much a compensation (e.g. Lohmann and Feichter, Atmos. Chem. Phys. 2006)

p2 l6: I think we should in general aim to be more specific about what we mean by "meteorology".

P2 l19: It would be useful to discuss Bender et al. (J Climate 2015 doi 10.1175/JCLI-D-15-0095.1)

P2 l27: It would be appropriate to cite Li et al. (Nature Geosci 2011, doi 10.1038/NGEO1313) here.

P2 l29: Again it would be good to specify what is meant by "meteorology"

p3 l13: abbreviate second as "s"

p3 l20: overcast at which scale (1 min≈ 600 m for 10 m/s wind speed?)

P2 l8: "s" instead of "second"

p4 l31: this is only true on climatological (monthly-mean) time scales (e.g. Nam and Quaas, Geophys. Res. Lettt., doi:10.1002/grl.50945)

p5 l2, l4: minute → min

p5 l12: the stricter criterion was second in the earlier sentence

p5 l18: This is an interesting result. Is the conclusion that clouds are independent of the turbulence and other boundary-layer properties, at least with regard to the LWP?

P5 l24: It would be useful to also list the other numbers: what is the fraction of data-points with f=1, what with f > 0.99?

P5 l27: To which extent are these two quantities independent at all? Is not actually one derived from the other one (Table 1)?

p8 l11: Why this choice? Why not choosing bins such that each contains the same amount of data?

P8 l17: But Fig. 3e still shows considerable f changes that dominate rCRE variability at low LWP.

P9 l1: Discussion perhaps on negative relation between $\tau$ and rCRE for lower LWP?

P9 l12: can one tease out the result more clearly e.g. by averaging over the LWP bins?

P9 l12: "this result suggests" → "this result confirms" (as this important point was discussed and explained earlier)

p9 l17: again, only on monthly timescales

p9 l25: as before "These results indicate" → "These results confirm"

p9 l31: good idea! How many points remain? Maybe show a joint histogram?

P9 l32: interesting result. What could be the interpretation?

P10 l11: for LWP bins? Or all data?

P10 l18: How can a conclusion about cloud-level aerosol conditions be drawn from these results?

P11 l12: RH measured where?

P11 l23: to have a physically more consistent equation, use the density of liquid water in the denominator (and then all quantities in SI units, or whatever consistent units)

p11 l26: typical for this type of scale and measurements, one should probably add and cite MdComiskey and Feingold (2012)

p14 l6: An obvious way to overcome the problem of variable LWP is to use droplet number concentration retrievals instead.

P15 l1: This conclusion seems only to stem from the examination of the three different effective radius retrievals. If this is intended as a main conclusion, more information about the retrievals is necessary, including some discussion on how reliable each one of these is.
p15 l4: "meteorological conditions" should better be explained. What actually is meant here is co-variability of the aerosol with cloud macroscopic quantities (LWP in particular), I believe.

P15 l7: "meteorological drivers", or rather liquid water path and cloud fraction?

p16 References Abbreviate journal names appropriately (many instances)
Barnard et al. (2008): correct journal name

p26 Figure 3: add a joint histogram perhaps?

p32 Figure 9d: one more tick mark on the x-axis to properly define it.

---

## Author Comment (AC1) · 23 Aug 2016

**Response to anonymous reviewer 1**

**1. Page 2, Line 4: Absorbing aerosol could also modify the atmospheric temperature profile and stability, and reduce cloud amount via the semi-direct effect (e.g., Koren et al., 2008). Koren et al. (2008) just provide the cloud amount change with aerosol optical depth but didn't show absorbing aerosol could modify the atmospheric stability. Huang et al. (2009) use the Fu-Liou radiation model and CERES radiation flux to derived the heating rate of aerosol layer and directly show the changes in temperature profile.**
**Huang J., Q. Fu, J. Su, Q. Tang, P. Minnis, Y. Hu, Y. Yi, and Q. Zhao, 2009: Taklimakan dust aerosol radiative heating derived from CALIPSO observations using the Fu-Liou radiation model with CERES constraints,Atmos. Chem. Phys., 9, 4011-4021.**
Answer: Thanks. This reference was included in the revised version of the manuscript.
We did look into aerosol extinction and single scattering albedo for a possible semi-direct effect but for the data considered aerosol amounts were simply to small to have an impact. This is now stated in the text.

**2. Authors used the aerosol index instead of CCN concentration in this study. As the SGP site do equipped with the Cloud Condensation Nuclei Particle Counter, why do not use this data directly?**
Answer: In this work 14-years of coincident measurements of aerosol, clouds and meteorological data were used. CCN concentration measurements were available at the SGP only from 2007 on. Furthermore, a complete scanning of CCN measurements over different supersaturation values takes about 1 hour. The use of CCN concentration would require the selection of a given supersaturation, which would considerably reduce the number of data points analysed. On the other hand, nephelometer light scattering measurements were available since 1998 at 1-minute resolution. To maximize the number of years included in the analysis, aerosol index (a quantity readily calculated from scattering measurements) was used as a proxy for CCN concentration. Shinozuka et al., 2015 propose a new methodology to estimate CCN (at a given supersaturation) using light scattering measurements. We have now used his approach in addition to Ai and the results are similar to the results previously obtained, as shown in Figures R1a-c, for a supersaturation of 0.6%. This new analysis does not result in changes to the main conclusions of this paper. The distribution of daily correlation between rCRE and CCN is centered at 0.02. Also, the scatter plot of the correlation between rCRE and CCN by the correlation of LWP and

CCN concentration shows a positive correlation of 0.42. These figures are included in the Supplementary section.

[Figure]

Figure R1: a) Relative cloud radiative effect (rCRE) as a function of liquid water path (LWP) colored by CCN concentration, b) daily distribution of the correlation between rCRE and CCN, and c) correlation between rCRE and CCN versus the correlation between LWP and CCN. To calculate CCN concentration, a supersaturation of 0.6% was considered.

**3. Authors mentioned that all of the relevant variables were averaged to 1-minute resolution. Does this time resolution is suitable for this study?**
Answer: The choice of 1-minute resolution resulted from a consideration of a number of factors such as a desire for representation of small-scale aerosol-cloud processes, large statistics, consideration of the temporal scales of variability of aerosol and cloud fields, and the recognition that averaging might be required to show clear trends. The 1-minute timescale balances these factors. A higher temporal resolution for cloud fields would have been more desirable for broken cloud fields but these are a small fraction of the current data set.

**4. Koren et al. (2008, ACP) show ω (550 hPa) and RH(350 hPa) yielded the highest correlations with the satellite-derived cloud properties, these parameters will be used to represent the primary meteorological controls on the cloud system. Herein, authors examined the impact of cloud macroscopic properties (fc and $\tau$ ) and meteorological variables (Di, LTS, and w'$^2$) on cloud radiative effect. How did authors choose those**

**variables?**

Answer: Cloud fraction and cloud optical depth were selected because they are the two main macroscopic cloud variables related to cloud radiative properties. $w'^2$ was used as a proxy for turbulence. Higher turbulence leads to an increase in vapor supersaturation, favoring the formation of cloud droplets. This effect could increase cloud albedo, also affecting cloud radiative forcing. Di is a meteorological index associated with the coupling between the surface and the boundary-layer. Di is an indicator of how well-mixed the atmosphere is below the cloud, and therefore, how well surface measurements of aerosol loading and properties represent cloud-level aerosol. LTS is another meteorological index associated with the strength of the inversion capping. This parameter is related to cloud fraction, a parameter that also influences cloud radiative forcing. A minor change was included in the text reinforcing the influence of turbulence on supersaturation.

---

## Author Comment (AC2) · 23 Aug 2016

**Response to reviewer 2 - Johannes Quaas**

**No measurements in $N_d$ are used in the study by Sena et al.. Rather, near-surface nephelometer measurements of the aerosol index are used as a proxy for cloud condensation nuclei concentrations (CCN) and subsequently for $N_d$. The results of the decomposition analysis to identify impacts of the various parameters on rCRE are presented in Fig. 3 where all three effects are convolved. More instructive still is Fig. 5 where only overcast cases are selected. At the theoretical level (Eq. 8 and 9), for a given LWP bin, rCRE can only be a (strictly monotonically increasing) function of $N_d$. In light of this, the results are puzzling. There is no clear relationship of rCRE at given LWP with near-surface aerosol index (Fig. 5a). There is no influence of w on rCRE (Fig. 5b). There is, however, a more systematic influence of the decoupling index and also of lower-tropospheric stability (LTS). The authors interpret that the aerosol impact is small. This is a straightforward interpretation of Figs. 5a and 6a. But how can this conclusion be true? Does this not imply the simple theoretical model in Eq. 8 and 9 is wrong? The other possible explanation is certainly that the nephelometer measurements near the surface are not a good proxy for in-cloud $N_d$. Perhaps one could test this by trying to relate remote sensing retrievals of $N_d$ to the aerosol index?**

Answer: The simple two-stream theoretical model from Eq. 8 is useful to provide insight into the expected behavior of rCRE with LWP and $N_d$. It shows that the impact of LWP on rCRE is much larger than the impact of $N_d$, based on the fact that cloud optical depth is proportional to $LWP^{5/6} N_d^{1/3}$; i.e., in a relative sense, cloud optical depth is 2.5 times more sensitive to LWP than it is to $N_d$. However, this simple model does not account for several different conditions experienced during an actual measurement, e.g., 3D radiative effects, near-cloud radiative absorption, changes in atmospheric stability, dry air entrainment and non-adiabatic processes. On the other hand, Figs. 5a and 6a are a result of 'actual' ground-based measurements of rCRE, LWP and Ai. They represent the bulk result of the interaction of all processes affecting cloud radiative properties. Eq. 8 represents a highly simplified system where 'all else' must be equal and uncertainty in the terms is not allowed.

The lack of coupling between aerosol concentrations at the surface and cloud base could also explain this result; this is addressed elsewhere in the paper and in the responses below.

Following the reviewer's suggestion, we have looked at ground-based remote sensing retrievals of Nd using column properties LWP and $\tau_c$. A clear trend is observed when plotting rCRE vs. LWP colored by $N_d$ (Figure R2). This clear aerosol influence is, however, a result of the fact that the $N_d$ and rCRE data are no longer independent due to the retrieval method. We have stressed in previous work that an independent estimate of $N_a$ and/or $N_d$ is critical for such studies.

[Figure]

Figure R2: rCRE as a function of LWP colored by Nd.

On the other hand, $A_i$ (or other CCN proxies, such as Shinozuka et al., 2015) are independent measurements of the aerosol that might affect cloud properties. They do not rely on the retrievals of macroscopic cloud properties themselves, as do $N_d$ retrievals. The decoupling index is an indicator of how well-mixed the atmosphere is, and therefore, how efficiently aerosol particles are transported to higher levels of the atmosphere. Therefore, low $D_i$ values could indicate conditions under which surface-based aerosol measurements represent cloud-level aerosol. To overcome this issue, in the manuscript the daily correlation between rCRE and $A_i$ was calculated for low and high $D_i$ values (Fig. S3). No differences were observed (both distributions where centered around 0). These results are supported by the findings of Delle Monache, 2004, referenced in the paper and discussed later in this response. In addition, Shinozuka's (2015) proxy was used to calculate CCN (see Figs. S1a-c in response to referee 1). To consider only well-coupled conditions, only low $D_i$ values were selected. Again, the conclusion didn't change. Under these conditions, the distribution of daily correlation between rCRE and CCN is centered at -0.04. The scatter plot of the correlation between rCRE and CCN by the correlation of LWP and CCN concentration shows a high positive correlation, of 0.57 (Figures R3a-b).

[Figure]

Figure R3: a) Daily distribution of the correlation between rCRE and CCN, and b) correlation between rCRE and CCN versus the correlation between LWP and CCN, for well-coupled conditions ($D_i < 0.25$). To calculate CCN concentration, a supersaturation of 0.6% was considered.

**A very interesting question is further, why is there the impact of decoupling index (Fig. 5c) and LTS (Fig. 5d), but not of w ? My own experience would point to spurious variability in LWP and cloud fraction that the binning into bins of 5 g m$^{-2}$ and the constraint of the retrieved cloud fraction at 100% is not able to completely inhibit. Given that despite the length of the data record not overly much data is available due to the conditioned sampling, an option would be to use coarser bins and see whether the effects are larger.**
Answer: Actually there is a weak trend of rCRE increasing with decreasing $w'^2$. This weak trend is associated with the types of clouds associated with each $w'^2$ range. Usually, broken-cumuli are associated with higher convection, therefore higher $w'^2$, and lower cloud fraction. On the other hand, stratiform-like clouds are associated with lower convection (lower $w'^2$), and higher cloud fraction. Lower (higher) cloud fraction leads to lower (higher) rCRE.
To understand the impact of spurious data, we have tried several LWP binning schemes, as suggested. We have found that changing the binning does not change the general behavior of the curve. Figure R4 shows an example of a different binning scheme, using 20 g/m$^2$ for the LWP bin.

[Figure]

Figure R4: rCRE by LWP colored by $w'^2$ (LWP bin: 20 g/m$^2$).

**Else it is possible that the retrievals of either rCRE or LWP somehow depend on decoupling index or on LTS, but I am not enough of an expert on the retrievals to say whether this is possible. Another possibility is that the diurnal cycle in the decoupling index (Fig. 12) impacts rCRE more strongly than one might anticipate via the not-eliminated impact of the solar zenith angle (Eq. 8).**

**It would be very useful if the authors could discuss these questions in order to understand to which extent the "negative" or "null" result of no influence of aerosols on rCRE is a robust finding.**

Answer: rCRE does depend on $D_i$ or LTS via the dependence of primary factors such as $f_c$ and LWP on $D_i$ and LTS but we see no reason why the *retrievals* of rCRE or LWP would be. The first algorithms used for LWP retrievals (Liljegren et al., 2001) could present some biases regarding $D_i$, as they used a statistical site-dependent approach, based on monthly coefficients dependent on near-surface temperature estimates. However, in this work, the MWR retrieval (MWRRET) value-added-product algorithm was used. This algorithm has been significantly improved and relies on physical retrievals of the temperature profile. Therefore, we don't see any reason for LWP retrievals to depend on $D_i$.

You are right: there is a non-eliminated relationship between rCRE and solar zenith angle on the two-stream theoretical model presented in Section 2. rCRE varies slowly with $\theta_0$ for lower $\theta_0$ values, but shows a strong dependence on $\theta_0$ for higher angles. Figure 3 provides useful information, but includes another degree of variability in the data ($\theta_0$), that prevents us from immediately attributing changes on rCRE to the other variables. Therefore, this intrinsic

dependence of rCRE on $\theta_0$ does not allow us to isolate the effects due solely to other properties on rCRE from the effects caused by solar illumination. To reduce this influence, another figure was included showing the dependence of rCRE on LWP colored by the same variables of Figure 3, but considering only cases obtained when $\cos(\theta_0) \geq 0.6$. This limit was selected such as to maximize the amount of data analyzed and at the same time, minimize the effects of solar illumination on rCRE. As $D_i$, $w'^2$ and $f_c$ are highly correlated and have a very marked diurnal cycle (and therefore an association with solar zenith angle) we cannot separate the impacts of Di (and fc) and solar zenith angle on rCRE on Figure 3. The new analysis, obtained after restricting the solar illumination angle, shows that the general trends of rCRE do not change for aerosol and $\tau_c$, when $\theta_0$ is limited. However, for $D_i$, $f_c$, $w'^2$ and LTS the rCRE trends at a fixed LWP value previously observed in these figures are reduced. One of the explanations for this behavior is that, as these variables have a marked diurnal cycle; limiting $\theta_0$ significantly reduces their variability. For example, higher $D_i$ values are usually observed during early-morning and late afternoon. Therefore when only low $\theta_0$ values are considered, these higher $D_i$ observations will not appear as frequently in the data set. On the other hand, as higher LWP values are associated with higher $f_c$, higher $D_i$ and lower $w'^2$ values, high rCRE values will likely be observed when these macroscopic properties and thermodynamic conditions are met. These points are now discussed in the manuscript. As this comment generated so many interesting and fruitful discussions we have decided to include both, the $\theta_0$-restricted and non-restricted figures in the manuscript (Figures 3 and 5 in the new version). For the other analysis ($A_c$ vs. LWP, daily distributions of $\rho_{rCRE,LWP}$ and $\rho_{rCRE,Ai}$, correlation between $\rho_{rCRE,Ai}$ and $\rho_{Ai,LWP}$) only the more restrictive condition for $\theta_0$ was used. It is worth mentioning that when we limit $\theta_0$, the daily correlations between rCRE and LWP increased significantly, and 98% of the cases show positive $\rho_{rCRE,LWP}$. Also, in the last figure the correlation between $\rho_{rCRE,Ai}$ and $\rho_{Ai,LWP}$ increases to 0.71. These results indicate that variations in $\theta_0$ might have been obscuring the relationships. The figure of the distributions of $\rho_{rCRE,Ai}$ for low and high Di conditions (Fig. 7 in the previous version of the manuscript) was removed, because when $\theta_0$ is restricted only low $D_i$ remain in the database.

**The authors also conclude that microphysical metrics in general are misleading. This conclusion mainly stems from the results in Table 2. While there is probably consensus that retrievals of cloud microphysical quantities are error-prone, such a broad conclusion should be corroborated better. If the authors decide this is a focus of the paper, then they should include a much more detailed discussion and ideally evaluation of the three different retrieval algorithms for the effective radius and explain why all three are (equally?) valid. In this regard, it should be noted that retrievals of microphysical quantities from the surface in general are more difficult than from the top of the atmosphere (Brueckner et al., J. Geophys. Res., 2014 doi: 10.1002/2014JD021775)**

Answer: Our intent is to focus the results of this paper on the influence of LWP and cloud macrophysical properties on rCRE in a statistical sense. Microphysical metrics, when used carefully, can likely provide a quantification of aerosol influences on cloud microphysical properties, and at the very least a test of self-consistency. However, transferring that value to a statistically representative rCRE is not as straightforward as the literature has assumed in the past. This issue is not the focus of the current paper but has been dealt with in some depth in earlier works (McComiskey et al. 2009; McComiskey and Feingold 2012).

We are quite interested in the results of Table 2 and plan to look further into these difference and better understand the retrieval uncertainties in general. Regarding the Brueckner reference we disagree that space based retrievals are in general easier than surface based retrievals, unless of course the surface microphysical retrievals are based on transmission, which then generates ambiguous results (Sebastian Schmidt and colleagues; Brueckner). None of our microphysical retrievals uses transmission as in those works so we feel it would be a distraction to engage in discussion of this topic. Given our focus on the current work, and our earlier efforts on ACI metrics, we do not feel that an in depth examination of retrieval uncertainties would fit within the current work.

**Besides these two main comments, I have a few specific comments listed below.**

**Specific comments**

**Title: Why not name the ARM SGP site?**
Answer: The title was modified and now mentions "the Southern Great Plains" explicitly.

**Abstract p1 l18: I think that in the abstract "weak" needs quantification. One would expect the aerosol to be second order anyway.**
Answer: The following sentence was added to the abstract: "On a daily basis, aerosol shows no correlation with cloud radiative properties (R = -0.01 $\pm$ 0.03) whereas liquid water path shows a positive correlation (R = 0.56 $\pm$ 0.02)".

**P1 l29: Most publications currently would suggest additional effects due to microphysical adjustments, not so much a compensation (e.g. Lohmann and Feichter, Atmos. Chem. Phys. 2006)**
Answer: In the original sentence we talk about both: "mutually compensating effects and adjustments". We don't think a change is necessary.

**p2 l6: I think we should in general aim to be more specific about what we mean by "meteorology".**
Answer: In order to be more specific, the sentence was modified to: "The influence of meteorological drivers and thermodynamic conditions (e.g., atmospheric stability and humidity) on aerosol-cloud interaction assessments is increasingly being brought into focus".

**P2 l19: It would be useful to discuss Bender et al. (J Climate 2015 doi 10.1175/JCLI-D-15-0095.1)**
Answer: Thanks for raising this. The main point of this paper is that for marine stratocumulus regimes, at a fixed cloud fraction, total albedo is controlled by temporal rather than spatial variability. This is a bit off topic and despite our efforts, we couldn't find a place to insert this idea without breaking the flow of the paper.

**P2 l27: It would be appropriate to cite Li et al. (Nature Geosci 2011, doi 10.1038/NGEO1313) here.**
Answer: We have modified our text to say: "The availability of such a large and comprehensive dataset provides an excellent opportunity to pursue a long-term study of the effects of aerosol and meteorology on the cloud radiative effect."

Li et al. addressed aerosol effects on precipitation. We have concerns about the way aerosol (CN rather than CCN), thermodynamics and macroscopic variables were taken into account in Li et al., 2011's paper and therefore chose not to cite it in the manuscript.

**P2 l29: Again it would be good to specify what is meant by "meteorology"**
Answer: The sentence was modified to "14-years of ground-based measurements at the SGP were analyzed to investigate the effects of aerosol and meteorological drivers (such as capping inversion strength, surface-boundary layer coupling and turbulence) on clouds".

**p3 l13: abbreviate second as "s"**
Answer: Done.

**p3 l20: overcast at which scale (1 min≈ 600 m for 10 m/s wind speed?)**
Answer: In this context, overcast conditions are considered on the scale of hundreds of meters. The average wind speed is around 6 m/s (1 min ~ 360 m). This information is now included in the revised manuscript.

**P2 l8: "s" instead of "second"**
Answer: Done.

**p4 l31: this is only true on climatological (monthly-mean) time scales (e.g. Nam and Quaas, Geophys. Res. Lettt., doi:10.1002/grl.50945)**
Answer: Some studies point out the relationship between LTS and $f_c$ on climatological (monthly-mean) time scales. However, Chen et al., Nature Geosc., 2014 point out the importance of LTS on cloud liquid water responses on a much shorter timeframe. In their work, instantaneous ECMWF reanalysis data are interpolated for each CloudSat cloud radar profile. This reference is now included in the newest version of the manuscript.

**p5 l2, l4: minute → min**
Answer: Done.

**p5 l12: the stricter criterion was second in the earlier sentence**
Answer: Thanks for noticing it. The order of the sentence has been inverted in the new version of the manuscript.

**p5 l18: This is an interesting result. Is the conclusion that clouds are independent of the turbulence and other boundary-layer properties, at least with regard to the LWP?**
Answer: We cannot affirm that clouds (or specifically LWP) are independent of these properties based only on the analysis of these histograms. Figure 1 only shows that the distributions of turbulence, $D_i$, LTS and $A_i$ do not depend significantly on the higher and lower end of the LWP distribution (30-50 $g/m^2$ and 150 - 250 $g/m^2$). However, to address the influence of these boundary-layer properties on clouds would require a much more rigorous analysis that delves into the full meteorological context of these data.

**P5 l24: It would be useful to also list the other numbers: what is the fraction of data-points**

**with f=1, what with f > 0.99?**
Answer: The fraction of data points with $f_c$=1 (or $f_c > 0.99$) is 79%, for LWP between 50 and 150 g/m$^2$ and 75% for LWP between 30 and 250 g/m$^2$. This information is now included in the revised manuscript.

**P5 l27: To which extent are these two quantities independent at all? Is not actually one derived from the other one (Table 1)?**
Answer: These quantities are not derived from each other, but they are closely related. According to Xie and Liu, 2013 the relation shown in equation (4) holds if we consider surface albedo = 0 and neglect cloud absorption of radiation. As usually surface albedo << 1 and cloud absorption is small, equation (4) is a good approximation for rCRE.

**p8 l11: Why this choice? Why not choosing bins such that each contains the same amount of data?**
Answer: The statistical distribution of LWP is asymmetrical and would typically require geometrically increasing bin width. However the lack of noise at the higher end of the LWP distribution suggests that there is no need to change to a geometrical bin structure. The binning choice used in the original manuscript makes the plot fairly symmetrical, using easy-to-read intervals (0.02 and 5 g/m$^2$, for rCRE and LWP, respectively). rCRE varies from 0 to 1, leading to 50 bin intervals. LWP varies from 0 to 250 g/m$^2$ (even though, only measurements with LWP > 30 g/m$^2$ were considered), also leading to 50 bin intervals. As shown above, we see no significant change in results for a different LWP bin width.

**P8 l17: But Fig. 3e still shows considerable f changes that dominate rCRE variability at low LWP.**
Answer: True. This is associated with the small amount of clouds with lower $f_c$ that bring down the average $f_c$ as explained in the text. The following sentence is now included in the manuscript: " Figure 3e shows considerable $f_c$ changes that dominate rCRE variability at low LWP." We believe the explanation was already in the text: "Some rCRE differences could be related to the relatively small number of broken cloud events that: i) reduce rCRE due to the smaller $f_c$ associated with this cloud type; and, ii) introduce the possibility of three-dimensional radiative effects (e.g., Wen et al. 2007), and therefore deviations from the simple two-stream model approximations that form the basis of the rCRE analysis".

**P9 l1: Discussion perhaps on negative relation between $\tau$ and rCRE for lower LWP?**
Answer: We don't see a negative relation between rCRE and tc at low LWP.

**P9 l12: can one tease out the result more clearly e.g. by averaging over the LWP bins?**
Answer: Figure R5 shows $w'^2$ averaged over LWP bins. For LWP < 100 g/m$^2$, $w'^2$ decreases with increasing LWP. This is mostly driven by the larger number of broken cumuli that have lower $f_c$, lower LWP and higher $w'^2$. As LWP increases, the number of broken cumuli in each LWP bin decreases, $f_c$ increases and $w'^2$ decreases. As LWP reaches higher values, almost no broken-cumuli are observed, the number of observations decreases (Fig. 1a) and therefore $w'^2$ saturates and becomes noisier.

[Figure]

Figure R5: $w'^2$ averaged according to LWP bins. The error bars represent the standard deviation of the mean value of $w'^2$.

**P9 l12: "this result suggests" → "this result confirms" (as this important point was discussed and explained earlier)**
Answer: Done.

**p9 l17: again, only on monthly timescales**
Answer: As previously pointed out, the impact of LTS on cloud liquid water responses was verified at a much shorter timeframe (Chen et al., Nature Geosc., 2014).

**p9 l25: as before "These results indicate" → "These results confirm"**
Answer: Done.

**p9 l31: good idea! How many points remain? Maybe show a joint histogram?**
Answer: Figure R6 shows the joint histogram of cloud albedo and LWP for overcast conditions ($f_c = 1$). This figure will be included in the Supplementary section.

[Figure]

Figure R6: Joint histogram of cloud albedo and LWP for fully overcast conditions ($f_c = 1$).

**P9 l32: interesting result. What could be the interpretation?**
Answer: This figure was removed in the new version of the manuscript. As previously pointed out by the referee and discussed here, rCRE increases with $\theta_0$. As high $D_i$ values are observed at times of the day when $\theta_0$ is high, due to its diurnal cycle, it is hard to attribute this this increase in rCRE solely to an increase in $D_i$. This increase in rCRE could be due to changes in $\theta_0$. The same analysis was performed using only data obtained when $\cos(\theta_0) \geq 0.6$. This new analysis does not show significant changes of $A_c$ with any of the analyzed variables. This discussion was modified in the new version of the manuscript.

**P10 l11: for LWP bins? Or all data?**
Answer: For this analysis all data with LWP between 50 and 150 g m$^{-2}$ were used. Cases that had less than 25 observations per day were excluded from the analysis.

**P10 l18: How can a conclusion about cloud-level aerosol conditions be drawn from these results?**
Answer: Under well-mixed (coupled) conditions surface-based measurements of aerosol would represent well cloud-level aerosol. In the old version of the manuscript Fig. 7 showed no significant difference between well-coupled and poorly-coupled conditions. Delle Monache et al., 2004 show that, at SGP, extensive and intensive aerosol properties measured at the surface and within the atmospheric boundary layer are well-correlated. Therefore we contend that at SGP surface-based measurements of aerosol properties are representative of the air within the atmospheric boundary-layer. In the new version of the manuscript this Figure and this excluded were removed, since only well-coupled conditions (low $D_i$) remained when solar zenith angle

was limited. However, as we consider Delle Monache's results a crucial point for the present analysis, the discussion on this paper was significantly extended in Section 2 of the new version of the manuscript.

**P11 l12: RH measured where?**
Answer: RH was measured at the surface. This information was included in the new version of the manuscript.

**P11 l23: to have a physically more consistent equation, use the density of liquid water in the denominator (and then all quantities in SI units, or whatever consistent units)**
Answer: The equation was modified according to the suggestion above.

**p11 l26: typical for this type of scale and measurements, one should probably add and cite McComiskey and Feingold (2012)**
Answer: This reference is now added to the revised manuscript.

**p14 l6: An obvious way to overcome the problem of variable LWP is to use droplet number concentration retrievals instead.**
Answer: As previously discussed, ground-based remote sensing retrievals of Nd are done using LWP and COD. Therefore, LWP, COD and $N_d$ retrievals are not independent.

**P15 l1: This conclusion seems only to stem from the examination of the three different effective radius retrievals. If this is intended as a main conclusion, more information about the retrievals is necessary, including some discussion on how reliable each one of these is.**
Answer: Even though this is an important finding, this is not the focus of this paper. Entering details and understanding all the intricacies involved in these retrievals and their uncertainties would require the development of a complete new work. Our experience with ACI metrics at SGP and our comparisons with independent work (Kim et al. JGR 2009; doi:10.1029/2003JD003721) has left us with significant concerns about the robustness of ACI retrievals. So while we only show a small sample of cases here, we are confident that the problem is endemic. When stating generally that microphysical metrics are not always reliable, we also consider the uncertainty presented in the literature concerning cloud microphysical properties from active remote sensing techniques.

**p15 l4: "meteorological conditions" should better be explained. What actually is meant here is co-variability of the aerosol with cloud macroscopic quantities (LWP in particular), I believe.**
Answer: True. The sentence has been modified accordingly in the manuscript.

**P15 l7: "meteorological drivers", or rather liquid water path and cloud fraction?**
Answer: You are right. We didn't mention the macroscopic cloud properties before. The sentence was modified to "relative effects of aerosols, macroscopic cloud properties and meteorological drivers". We decided to maintain the expression "meteorological drivers", as we also analyzed variables related to turbulence, capping inversion and atmosphere-surface coupling.

**p16 References Abbreviate journal names appropriately (many instances) Barnard et al. (2008): correct journal name**

Answer: Done.

**p26 Figure 3: add a joint histogram perhaps?**
Answer: Figure R7 shows the joint histogram of rCRE and LWP. This figure will be included in the Supplementary section of the manuscript.

[Figure]

Figure R7: Joint histogram of rCRE and LWP.

**p32 Figure 9d: one more tick mark on the x-axis to properly define it.**
Answer: Done.